# Predicting Global Label Relationship Matrix for Graph Neural Networks under Heterophily

**Langzhang Liang**
Harbin Institute of Technology, Shenzhen
lazylzliang@gmail.com

**Xiangjing Hu**
Harbin Institute of Technology, Shenzhen
starry.hxj@gmail.com

**Zenglin Xu**[*]
Harbin Institute of Technology, Shenzhen & Pengcheng Lab
zenglin@gmail.com

**Zixing Song**
The Chinese University of Hong Kong
zxsong@cse.cuhk.edu.hk

**Irwin King**
The Chinese University of Hong Kong
king@cse.cuhk.edu.hk

## Abstract

Graph Neural Networks (GNNs) have been shown to achieve remarkable performance on node classification tasks by exploiting both graph structures and node features. The majority of existing GNNs rely on the implicit homophily assumption. Recent studies have demonstrated that GNNs may struggle to model heterophilous graphs where nodes with different labels are more likely connected. To address this issue, we propose a generic GNN applicable to both homophilous and heterophilous graphs, namely Low-Rank Graph Neural Network (LRGNN). Our analysis demonstrates that a signed graph's global label relationship matrix has a low rank. This insight inspires us to predict the label relationship matrix by solving a robust low-rank matrix approximation problem, as prior research has proven that low-rank approximation could achieve perfect recovery under certain conditions. The experimental results reveal that the solution bears a strong resemblance to the label relationship matrix, presenting two advantages for graph modeling: a block diagonal structure and varying distributions of within-class and between-class entries.

## 1 Introduction

Graphs (or networks) are ubiquitous in various fields, such as social networks [Tang et al., 2013, Xu et al., 2015, 2019b], biology [Guzzi and Zitnik, 2022], and chemistry [Gilmer et al., 2017]. Many real-world networks follow the Homophily assumption, *i.e.*, linked nodes tend to share the same label or have similar features; while for graphs with heterophily, nodes with different labels are more likely to form a link. For example, many people tend to connect with people of the opposite sex in dating networks. Graph Neural Networks (GNNs) [Kipf and Welling, 2017, Velickovic et al., 2018] have shown significant success in tackling a diverse set of graph mining tasks, such as NLP [Song and King, 2022, Chen et al., 2023] and clustering [Huang et al., 2023, Kang et al., 2020]. There are also studies dedicated to enhancing the representational power of GNNs [Liang et al., 2023, Xu et al., 2019a]. Recently, researchers have observed that GNNs may face difficulties in learning on graphs

---

[*]Corresponding author.

37th Conference on Neural Information Processing Systems (NeurIPS 2023).

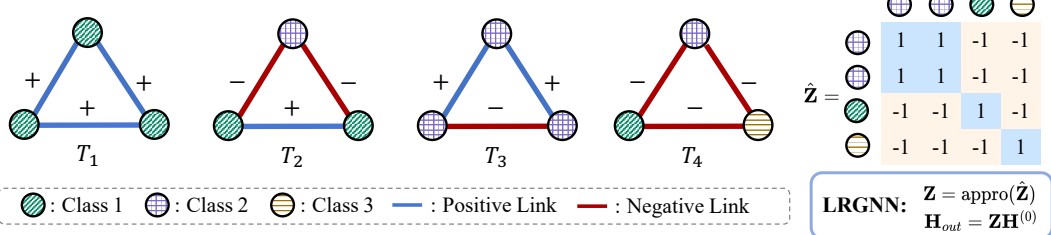

Figure 1: Undirected signed triads. Structural balance theory states that $T_1$ and $T_2$ are balanced, whereas $T_3$ and $T_4$ are unbalanced. Weak balance states that only $T_3$ is unbalanced. Clearly, triads of signed graphs obey weak balance. Consequently, the label relationship matrix $\hat{\mathbf{Z}}$ exhibits a low-rank structure. Then $\hat{\mathbf{Z}}$ can be recovered by low-rank approximation.

with heterophily due to the smoothing operation, which tends to generate similar node representations for adjacent nodes, even though their labels differ [Abu-El-Haija et al., 2019, Zhu et al., 2020].

Various designs [Zhu et al., 2020, Chien et al., 2021, Lim et al., 2021] have been proposed to enhance the performance of GNNs under heterophilous scenarios (see Zheng et al. [2022] for a survey). Among them, high-pass filters are the most frequently used components as they can push away a node from its neighbors in the embedding space, aligning with the characteristic of heterophily. Spectral-based methods [Chien et al., 2021, Luan et al., 2021] combine high-pass filters with low-pass ones by fusing the outputs from intermediate layers. However, these methods cannot capture the node-level homophily ratio, as they use only one convolutional filter type in each layer. Alternatively, some methods [Bo et al., 2021, Yang et al., 2021] enable neural networks to learn the aggregation coefficients. Specifically, these methods update node representation by computing a learned signed weighted combination of neighbors' representations by adopting graph attention function [Velickovic et al., 2018]. However, the graph attention function has been proven to compute a form of static attention: the ranking of attention scores remains unconditioned on the query [Brody et al., 2022].

According to Zhu et al. [2020], under specific conditions, a node's 2-hop neighborhood is expected to be predominantly homophilic. To capitalize on this result, they suggest amplifying a node's influence using its 2-hop neighbors. Building on this, GloGNN [Li et al., 2022] introduces a subspace clustering approach to derive a global coefficient matrix. Nonetheless, Liu et al. [2010] proves that this matrix's between-class entries are almost zero. This indicates that while GloGNN taps into global homophily, it rarely employs remote heterophilous nodes for feature propagation. Moreover, the subspace clustering approach ensures dense within-class matrix entries but does not account for the sign of these entries.

In this paper, we aim at *recovering a global label relationship matrix $\hat{\mathbf{Z}}$ such that $\hat{\mathbf{Z}}_{i,j} = 1$ if node $v_i$ and $v_j$ have the same label (homophilous) and $-1$ otherwise (heterophilous)*. In this case, we exploit global homophily and heterophily by assigning proper signs. Determining $\hat{\mathbf{Z}}_{i,j}$, which requires knowledge of node labels, seems impractical due to the unavailability of many labels during training. Here a question arises: can we infer the unknown $\hat{\mathbf{Z}}_{i,j}$ from a small set of observations?

Unlike previous neural network-based methods, we employ sign inference to solve the problem. Signed social networks (SSNs) utilize positive links to denote friendship between two users and negative links to represent enmity. Sign inference is the task of predicting the signs of unknown links using the knowledge of available ones. The structural balance theory [Cartwright and Harary, 1956], a fundamental principle of SSN, postulates that individuals in signed networks are disposed to following specific patterns of "an enemy of my friend is my enemy" (e-f-e), "a friend of my friend is my friend" (f-f-f), "a friend of my enemy is my enemy" (f-e-e), and "an enemy of my enemy is my friend" (e-e-f). However, the last assumption could contradict actual scenarios. Therefore, a more general notion, called weak balance, was introduced to address this issue [Davis, 1967].

We study signed graphs, wherein a negative edge signifies that the two nodes possess distinct labels. Figure 1 illustrates that triads in signed graphs follow the weak balance theory, which suggests that the signed graph has a global low-rank structure [Hsieh et al., 2012]. An intriguing finding [Candès and Tao, 2010] demonstrates that if a matrix has a low rank, it can be accurately recovered from a

limited number of observations. Motivated by this, we approximate $\hat{\mathbf{Z}}$ with the solution to a low-rank matrix approximation problem, which is then used for feature propagation.

Since the solution is a low-rank matrix, our new model is termed Low-Rank Graph Neural Network (LRGNN). We summarize the contributions of this paper as:

- We show that triads in signed graphs obey weak balance, based on which the unknown label relationship between two nodes can be inferred by looking up the known ones.
- We prove that a variant of signed GNNs shows a tendency to follow the structural balance theory, which can be enhanced by eliminating the faulty assumption from the model design.
- We propose an effective model LRGNN. Using both real-world and synthetic datasets, we provide comprehensive experimental results, thereby highlighting the superior performance of LRGNN over other state-of-the-art methods.

## 2   Preliminaries

We first introduce the notations used in this paper.

**Notations**. Denote by $\mathcal{G} = (\mathcal{V}, \mathcal{E})$ an undirected graph, where $\mathcal{V}$ and $\mathcal{E}$ denote the node set and edge set, respectively. The nodes are described by a node feature matrix $\mathbf{X} \in \mathbb{R}^{n \times f}$, where $n$ and $f$ are the number of nodes and number of features per node, respectively. $\mathbf{Y} \in \mathbb{R}^{n \times c}$ is the node label matrix. The neighbor set of node $v_i$ is denoted by $\mathcal{N}_i$. Let $\mathbf{A} \in \mathbb{R}^{n \times n}$ denote the adjacency matrix where $\mathbf{A}_{i,j} = 1$ if $(i, j) \in \mathcal{E}$ and 0 otherwise. We define a label relationship matrix $\hat{\mathbf{Z}} \in \mathbb{R}^{n \times n}$ for a signed graph, where $\hat{\mathbf{Z}}_{i,j} = 1$ if $(i, j)$ share the same label and $-1$ otherwise. Let $\tilde{\mathbf{A}}$ be a signed adjacency matrix indicated by an observation set $\Omega$, with $\tilde{\mathbf{A}}_{i,j} = \hat{\mathbf{Z}}_{i,j}$ if $(i, j) \in \Omega$ and 0 otherwise. The Frobenius norm of a matrix is given by $\| \cdot \|_F^2$. $\mathbf{A}_{i,:}$ represents the $i$-th row of matrix $\mathbf{A}$.

### 2.1   Link Sign Prediction via Low-Rank Matrix Completion

Next, we discuss the problem of link sign prediction and the theoretical support for using low-rank matrix completion.

**Link Sign Prediction.**   The link sign prediction task involves determining the signs of unobserved links in a network, where some links are already labeled as positive or negative. This work specifically considers scenarios where only one link exists between each node pair $(i, j) \in \mathcal{V} \times \mathcal{V}$. To be precise, *given $\tilde{\mathbf{A}}$ partly observed from $\hat{\mathbf{Z}}$, the objective is to predict the signs of the unknown entries of $\hat{\mathbf{Z}}$.* Note that the observed entries can be constructed using available labels from the training set. This task is difficult in general because of the low supervised ratio. For instance, if the supervised ratio is $30\%$, we need to guess $91\%$ of entries of $\hat{\mathbf{Z}}$ using the knowledge of the rest $9\%$ of entries.

Here is a fact that significantly changes the task's premise, rendering it practical. The rank of $\hat{\mathbf{Z}}$ is equivalent to the number of classes if $c \geq 2$. This low-rank structure is described by the weak balance theory: for homophilous node pair $(i, j)$ and any $v_k \in \mathcal{V}$, if $\hat{\mathbf{Z}}_{i,k} = 1$ then $\hat{\mathbf{Z}}_{j,k} = 1$ (f-f-f), if $\hat{\mathbf{Z}}_{i,k} = -1$ then $\hat{\mathbf{Z}}_{j,k} = -1$ (f-e-e); for heterophilous $(i, j)$, if $\hat{\mathbf{Z}}_{i,k} = 1$ then $\hat{\mathbf{Z}}_{j,k} = -1$ (e-f-e). Therefore, $\hat{\mathbf{Z}}$ has $c$ linearly independent distinct rows if $c \geq 2$ (rank($\hat{\mathbf{Z}}$) $= c$). Inspired by this finding, we employ *Matrix Completion* to address this challenge.

**Matrix Completion.**   The task of matrix completion involves predicting the missing values in an unknown matrix based on a limited number of observed entries in the observation set $\Omega$. Exact recovery of a matrix from a small $\Omega$ is impossible without making any assumption about the matrix. But the search for solutions becomes meaningful as soon as the unknown matrix has a low rank; numerous algorithms can accomplish a near-perfect recovery with high probability under mild assumptions. This is known as the Low-Rank Matrix Completion (LRMC) problem.

**Low-Rank Matrix Completion.**   Given a signed adjacency matrix $\tilde{\mathbf{A}}$ and the observation set $\Omega$, the task of LRMC is to find the lowest-rank solution among all the feasible solutions as follows,

$$\min \quad \text{rank}(\mathbf{Z}), \quad s.t. \quad \mathrm{P}_\Omega(\mathbf{Z}) = \mathrm{P}_\Omega(\tilde{\mathbf{A}}), \tag{1}$$

where $\mathbf{Z}$ is the decision variable and $\mathrm{P}_\Omega(\cdot)$ is the projection onto the observation set. Unfortunately, (1) is an NP-hard problem without known algorithms capable of solving problems in practical time [Candès and Tao, 2010]. A matrix of rank $r$ has exactly $r$ nonzero singular values. Hence, the rank function is simply the number of nonvanishing singular values. As the $l_1$-minimization is the tightest convex relaxation of the combinatorial $l_0$-minimization problem [Candès and Plan, 2010]. An alternative is the nuclear minimization problem,

$$\min \quad \|\mathbf{Z}\|_*, \quad s.t. \quad \mathrm{P}_\Omega(\mathbf{Z}) = \mathrm{P}_\Omega(\tilde{\mathbf{A}}), \tag{2}$$

where $\|\mathbf{Z}\|_*$ denotes the nuclear norm of $\mathbf{Z}$, which is the sum of the singular values. Equivalently we can reformulate Eq.(2) in Lagrange form

$$\min \quad \frac{1}{2}\|\mathrm{P}_\Omega(\mathbf{Z} - \tilde{\mathbf{A}})\|_F^2 + \lambda\|\mathbf{Z}\|_*, \tag{3}$$

where $\lambda \geq 0$ is a regularization parameter. A surprising result involving LRMC is that under certain assumptions, the missing entries of $\tilde{\mathbf{A}}$ can be accurately predicted.

*Theorem* 1. [Candès and Tao, 2010] Let $\hat{\mathbf{Z}} \in \mathbb{R}^{n \times n}$ be a fixed matrix of rank $r = O(1)$ obeying the strong incoherence property with parameter $\varepsilon$. Suppose we observe $m$ entries of $\hat{\mathbf{Z}}$ with locations sampled uniformly at random. Then there is a positive numerical constant $C$ such that if

$$m \geq C\varepsilon^4 n(\log n)^2 \tag{4}$$

then $\hat{\mathbf{Z}}$ is the unique solution to Eq.(2) with probability at least $1 - n^{-3}$. In other words: with high probability, nuclear-norm minimization recovers all the entries of $\hat{\mathbf{Z}}$ with no error.

Solving Eq.(2) requires the computation of a SVD of a potentially large matrix, which can be computationally expensive. To reduce this cost, we can use an SVD-free low-rank matrix factorization (LRMF) problem formulated as follows:

$$\min_{\mathbf{U},\mathbf{V} \in \mathbb{R}^{n \times q}} \quad \|\mathrm{P}_\Omega(\mathbf{U}\mathbf{V}^T - \tilde{\mathbf{A}})\|_F^2 + \lambda(\|\mathbf{U}\|_F^2 + \|\mathbf{V}\|_F^2), \tag{5}$$

where $c \leq q \ll n$ is the operating rank. There is a remarkable fact that ties the LRMF problem and the nuclear-norm regularized problem (3) [Mazumder et al., 2010]: by selecting a $q \geq c$, the solution to Eq.(5) also provides a solution to Eq.(3). The equivalence between the solutions allows us to approximate $\hat{\mathbf{Z}}$ by the solution to Eq.(5).

We present some fundamental facts that support the recovery of $\hat{\mathbf{Z}}$. The first fact is that the rank of $\hat{\mathbf{Z}}$ is deterministic. Ideally, $q = rank(\hat{\mathbf{Z}})$ should be the optimal choice [Hastie et al., 2015]. The second fact is that $\hat{\mathbf{Z}}_{i,;} = \hat{\mathbf{Z}}_{j,:}$ holds for any homophilous $(i,j)$. This implies that class imbalance is the upper bound of the coherence parameter $\varepsilon$. In most cases, $\varepsilon = O(1)$. Furthermore, if $\tilde{\mathbf{A}}_{i,j} = 1$ is observed, $v_i$ and $v_j$ can combine their observations. For example, if $\tilde{\mathbf{A}}_{i,k}$ is observed, then $\tilde{\mathbf{A}}_{j,k}$ is also ascertainable, regardless of whether it is directly observed or not ($\tilde{\mathbf{A}}_{i,k} = \tilde{\mathbf{A}}_{j,k}$). As a result, there is a significant increase in the observation rate.

In real-life data, we typically witness a deviation from the assumption that observations exhibit uniform distribution. Nonetheless, this deviation only moderately influences the performance. Suppose a node lacks any observed samples. In that case, an exact recovery is theoretically impossible, but it would not hinder the recovery of other rows. Low-rank approximation methods have been applied to practical applications with a fair amount of success, such as the Netflix problem [Koren et al., 2009].

## 3 Structural Balance Theory and Signed GNNs

We consider a variant of signed GNNs [Bo et al., 2021] that updates node representations as

$$h_i^{(l+1)} = \sum_{j \in \mathcal{N}(i)} \alpha_{i,j}^{(l)} h_j^{(l)}, \quad \alpha_{i,j} = f_\theta(h_i^{(l)}, h_j^{(l)}), \tag{6}$$

where $f_\theta(\cdot, \cdot)$ refers to a neural network that measures the similarity between the inputs and returns a scalar ranging in $[-1, 1]$. Let $y_{i,j} = 1$ if two neighboring nodes $v_i$ and $v_j$ belong to the same class and $y_{i,j} = -1$ otherwise. Let $< i, j, k >$ be a triad such that $v_j$ is a neighbor of $v_i$ and $v_k$ is a neighbor of $v_j$, while $v_i$ and $v_k$ are not directly linked. We have the following result.

*Theorem* 2. Consider that we apply a $L$-layer signed GNN ($L \geq 2$) on a triad. The output of the signed GNN is equivalent to $\mathbf{H}^{(L)} = \mathbf{Z}\mathbf{H}^{(0)}$, where $\mathbf{Z}$ is a matrix. Assume that each coefficient $\alpha_{i,j}$ is independent of other coefficients and the probability that the model can precisely predict the sign of each coefficient is $p$, namely $Pr(\text{sign}(\alpha_{i,j}^{(l)}) = y_{i,j}) = p$. Also, assume that all the self-coefficients are positive. The probability that at least one of $\mathbf{Z}_{i,j}$ and $\mathbf{Z}_{j,k}$ are correct in sign and the product $\mathbf{Z}_{i,j}\mathbf{Z}_{j,k}\mathbf{Z}_{i,k} > 0$ is given by $p_b$. Then $p_b$ is monotonically increasing concerning $p$. Especially, if $p = 1$, $\mathbf{Z}_{i,j}\mathbf{Z}_{j,k}\mathbf{Z}_{i,k} > 0$ always holds.

*Remark* 1. A triad is said to be balanced iff its product is positive as the four types of triads given by balance theory have two or zero negative signs.

The above result suggests that multi-layer signed GNNs may implicitly implement the structural balance theory as they frequently create balanced triads. When p equals 1, all triads are balanced, indicating that the model is following the idea that "a heterophilous neighbor of my heterophilous neighbor is a homophilous node." Nonetheless, this notion does not necessarily hold under multi-class classification tasks. Hence, even if the model can predict the label relationship between direct neighbors accurately, it will still likely predict the relationship between 2-hop neighbors inaccurately. This suggests that multiple-layer signed GNNs are sub-optimal. Signed Graph Convolutional Networks (SignedGNNs) [Derr et al., 2018] utilizes the balance theory to model signed graphs. By eliminating the assumption "e-e-f" (see Supplement for implementation details), we obtain a weakly balanced version of SignedGNNs termed WB-SignedGNNs. To evaluate their vulnerability to over-smoothing, we compared the performance of WB-SignedGCN and SignedGCN at different depths, ranging from 4 to 24. As Figure 2 illustrates, SignedGCN suffer from over-smoothing, and this problem can be largely resolved by dropping the incorrect assumption in the model's design. Such a minute adjustment in the design can result in substantial performance enhancement for 24-layer SignedGCN (80% vs. 53% on Wisconsin dataset).

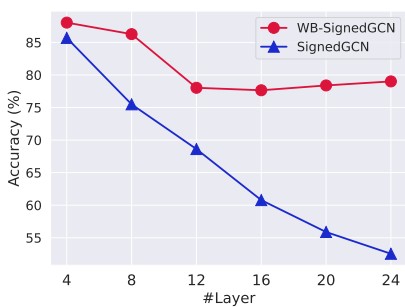

Figure 2: Over-smoothing analysis.

## 4   Approach

In this section, we present the overall framework of the Low-Rank Graph Neural Networks.

### 4.1   Predicting the label relationship matrix via robust low-rank approximation

Note that in $\hat{\mathbf{Z}}$, the information on the original graph topology is lost: we cannot tell whether two given nodes are linked or not since all entries are nonzero. To preserve the graph topology information, following Lim et al. [2021], we first apply MLPs to fuse feature matrix and adjacency matrix into a lower-dimensional matrix $\mathbf{H}^{(0)} \in \mathbb{R}^{n \times c}$,

$$\mathbf{H}^{(0)} = (1 - \mu)MLP_X(\mathbf{X}) + \mu MLP_A(\mathbf{A}), \quad (7)$$

where $0 < \mu < 1$ is the balance term. Next, we utilize LRMF to recover $\hat{\mathbf{Z}}$ with the edge set being the observation set ($\Omega = \mathcal{E}$), matching the time complexity of vanilla GCNs [Kipf and Welling, 2017]. Furthermore, an attention term [Bahdanau et al., 2015] is incorporated into Eq.(5) as a supplement to the matrix completion. We approximate $\hat{\mathbf{Z}}$ with $\mathbf{Z} = \mathbf{U}_* \mathbf{V}_*^T$ by solving the following robust low-rank matrix factorization problem,

$$F_c(\mathbf{U}, \mathbf{V}) = \sum_{(i,j) \in \mathcal{E}} \min(((\mathbf{U}\mathbf{V}^T)_{i,j} - \tilde{\mathbf{A}}_{i,j})^2, \tau_{i,j}) + \lambda\|\mathbf{U}\|_F^2 + \lambda\|\mathbf{V}\|_F^2 +$$
$$\gamma \sum_{i,j}(h_i^{(0)^T}\mathbf{W}h_j^{(0)} - (\mathbf{U}\mathbf{V}^T)_{i,j})^2, \quad (8)$$

where $\lambda > 0$ and $\gamma > 0$ are hyper-parameters, $\mathbf{W}$ a parameter matrix, $h_i^{(0)^T}\mathbf{W}h_j^{(0)}$ an attention coefficient, $\tau_{i,j}$ a parameter of the capped norm. Low-rank matrix approximation models are prone

to outliers due to the square loss function [Keshavan et al., 2010]. To mitigate this issue, this work applies capped norm to limit the contribution of the approximation error $((\mathbf{U}\mathbf{V}^T)_{i,j} - \tilde{\mathbf{A}}_{i,j})^2$. This strategy has been extensively utilized by researchers [Nie et al., 2014, 2017, Jiang et al., 2015]. Note that here each sample $(i,j)$ has an independent parameter $\tau_{i,j}$ which will be explained in detail later. The min operator render the function non-differentiable, and the projection function prevents us from obtaining a closed-form solution. It seems that solving Eq.(8) is infeasible. Using the surrogate function and Majorization-Minimization algorithm [Hastie et al., 2015], we can ensure that the objective function does not increase after each iteration. The optimization algorithm, initialization method, and proof of convergence can be found in the Supplement.

Once we have derived $\mathbf{U}_*\mathbf{V}_*^T$ by minimizing Eq.(8), the output of the model is formulated as

$$\mathbf{H}_{out} = ((1 - \beta)\mathbf{Z} + \beta\mathbf{I}_n)\mathbf{H}^{(0)}, \tag{9}$$

where the identity matrix $\mathbf{I}_n$ is used to augment the ego-information and $0 < \beta < 1$ is a hyper-parameter. A 1-layer network is sufficient to aggregate all nodes' representations for a node, it also averts the implicit usage of the balance theory.

*Remark* 2. The time complexity of LRGNN is proportional to the number of edges. A detailed discussion can be found in the Supplement.

## 4.2 Parameterized outlier detection

This section outlines a practical algorithm for detecting outliers based on the reliability of the estimated $\tilde{\mathbf{A}}_{i,j}$. To generate the signed adjacency matrix $\tilde{\mathbf{A}}$, we can use any off-the-shelf neural network classifier to generate pseudo labels. Additionally, the known node labels in training set $\mathcal{T}_\mathcal{V}$ and label matrix $\mathbf{Y}$ can also be utilized. Similar to Zhu et al. [2021], the pseudo labels are generated as follows,

$$\bar{\mathbf{Y}} = \mathbf{O} \odot \mathbf{Y} + (1 - \mathbf{O}) \odot \hat{\mathbf{Y}}, \ \hat{\mathbf{Y}} = \text{softmax}(\mathbf{P}), \ \mathbf{P} = f_{NN}(\mathbf{X}), \tag{10}$$

where $f_{NN}(\cdot)$ denotes a trained neural network named estimator, $\odot$ the Hadamard product, and $\mathbf{O}_{i,:} = \mathbf{1}$ if $i \in \mathcal{T}_\mathcal{V}$ and $\mathbf{0}$ otherwise. The signed adjacency matrix is defined as

$$\tilde{\mathbf{A}}_{i,j} = \begin{cases} \langle \bar{\mathbf{Y}}_{i,:}, \bar{\mathbf{Y}}_{j,:} \rangle - \delta & (i,j) \in \mathcal{E} \\ 0 & \text{otherwise} \end{cases} \tag{11}$$

Here $0 < \delta < 1$ is a parameter that controls the ratio of negative edge weights. $\langle \cdot, \cdot \rangle$ denotes the inner product of two vectors. Although this strategy appears simple, it is highly effective. The critical point here is that we do not need to know the exact classes of the two nodes and we just care about whether they belong to the same class. Let $p$ be the accuracy of the estimator and assume that there is a uniform distribution chance of being any incorrect class, namely $\frac{1-p}{c-1}$. The estimator has a probability of $\frac{(1-p)(pc+c-2)}{(c-1)^2}$ to identify any two heterophilous nodes as homophilous nodes. This probability is less than $1 - p$, provided that $c \geq 3$ and $p < 1$. For instance, when the accuracy of the estimator is $50\%$ and $c = 5$, for heterophilous $(i,j)$, the probability of assigning them different labels is about $82.8\%$, which is significantly higher than $50\%$. In our experiments, we use simple estimators including GCN and MLP for fairness.

| | Edge Hom. | #Nodes | #Edges | #Features | #Classes |
|---|---|---|---|---|---|
| **Texas** | 0.21 | 183 | 295 | 1,703 | 5 |
| **Wisconsin** | 0.11 | 251 | 466 | 1,703 | 5 |
| **Cornell** | 0.30 | 183 | 280 | 1,703 | 5 |
| **Actor** | 0.22 | 7,600 | 26,752 | 931 | 5 |
| **Squirrel** | 0.22 | 5,201 | 198,493 | 2,089 | 5 |
| **Chameleon** | 0.23 | 2,277 | 31,421 | 2,325 | 5 |
| **Cora** | 0.81 | 2,708 | 5,278 | 1,433 | 6 |
| **Citeseer** | 0.74 | 3,327 | 4,676 | 3,703 | 7 |
| **Pubmed** | 0.80 | 19,717 | 44,327 | 500 | 3 |
| **Penn94** | 0.47 | 41,554 | 1,362,229 | 5 | 2 |
| **arXiv-year** | 0.22 | 169,343 | 1,166,243 | 128 | 5 |
| **genius** | 0.61 | 421,961 | 984,979 | 12 | 2 |

Table 1: Dataset statistics.

The succeeding section discusses the design of $\tau_{i,j}$, which can be expressed as $\tau_{i,j} = c_{i,j} \cdot \tau$, where $0 < c_{i,j} < 1$ is a parameter. We adopt a widely accepted definition of outliers [Yeh, 2007],

$$\tau = q_3 + 1.5 \times (q_3 - q_1), \tag{12}$$

| | Texas | Wiscon. | Cornell | Actor | Squir. | Chamel. | Cora | Citeseer | Pubmed |
|---|---|---|---|---|---|---|---|---|---|
| MLP | 80.81±4.75 | 85.29±3.31 | 81.89±6.40 | 36.53±0.70 | 28.77±1.56 | 46.21±2.99 | 75.69±2.00 | 74.02±1.90 | 87.16±0.37 |
| GCN | 55.14±5.16 | 51.76±3.06 | 60.54±5.30 | 27.32±1.10 | 53.43±2.01 | 64.82±2.24 | 86.98±1.27 | 76.50±1.36 | 88.42±0.50 |
| GAT | 52.16±6.63 | 49.41±4.09 | 61.89±5.05 | 27.44±0.89 | 40.72±1.55 | 60.26±2.50 | 87.30±1.10 | 76.55±1.23 | 86.33±0.48 |
| H$_2$GCN | 84.86±7.23 | 87.65±4.98 | 82.70±5.28 | 35.70±1.00 | 36.48±1.86 | 60.11±2.15 | 87.87±1.20 | 77.11±1.57 | 89.49±0.38 |
| GPR-GNN | 78.38±4.36 | 82.94±4.21 | 80.27±8.11 | 34.63±1.22 | 31.61±1.24 | 46.58±1.71 | 87.95±1.18 | 77.13±1.67 | 87.54±0.38 |
| WRGAT | 83.62±5.50 | 86.98±3.78 | 81.62±3.90 | 36.53±0.77 | 48.85±0.78 | 65.24±0.87 | 88.20±2.26 | 76.81±1.89 | 88.52±0.92 |
| GloGNN++ | 84.05±4.90 | 88.04±3.22 | 85.95±5.10 | 37.70±1.40 | 57.88±1.76 | 71.21±1.84 | 88.33±1.09 | 77.22±1.78 | 89.24±0.39 |
| GGCN | 84.86±4.55 | 86.86±3.29 | 85.68±6.63 | 37.54±1.56 | 55.17±1.58 | 71.14±1.84 | 87.95±1.05 | 77.14±1.45 | 89.15±0.37 |
| ACM-GCN | 87.84±4.40 | **88.43±3.22** | 85.14±6.07 | 36.28±1.09 | 54.40±1.88 | 66.93±1.85 | 87.91±0.95 | 77.32±1.70 | 90.00±0.52 |
| LINKX | 74.60±8.37 | 75.49±5.72 | 77.84±5.81 | 36.10±1.55 | 61.81±1.80 | 68.42±1.38 | 84.64±1.13 | 73.19±0.99 | 87.86±0.77 |
| OGNN | 86.22±4.12 | 88.04±3.63 | **87.03±4.73** | **37.99±1.00** | 62.44±1.96 | 72.28±2.29 | **88.37±0.75** | 77.31±1.73 | 90.15±0.38 |
| **LRGNN** | **90.27±4.49** | 88.23±3.54 | 86.48±5.65 | 37.34±1.78 | **74.38±1.96** | **79.16±2.05** | 88.33±0.89 | **77.53±1.31** | **90.24±0.64** |

Table 2: Node classification accuracy (%) on small real-world benchmark datasets. The best results are highlighted in bold, whereas runner-up results are underlined. Each experiment is executed 10 times.

| | GCN | GAT | GPR-GNN | GloGNN++ | ACM-GCN | LINKX | OGNN | **LRGNN** |
|---|---|---|---|---|---|---|---|---|
| **Penn94** | 82.47±0.27 | 81.53 ±0.55 | 81.38±0.16 | 85.74±0.42 | 82.52±0.96 | 84.71±0.52 | 83.31±0.54 | **86.48 ± 0.52** |
| **arXiv-year** | 46.02±0.26 | 46.05±0.51 | 45.07±0.21 | 54.79±0.25 | 47.37±0.59 | **56.00±1.34** | 54.49±0.29 | 55.68 ± 0.35 |
| **genius** | 87.42±0.37 | 55.80±0.87 | 90.05±0.31 | 90.91±0.13 | 80.33±3.91 | 90.77±0.27 | 88.52±0.45 | **91.13 ± 0.12** |

Table 3: Node classification accuracy (%) on large-scale datasets.

where $q_1$ and $q_3$ are the first and third quartiles, respectively. Mathematically, $c_{i,j}$ is defined as

$$c_{i,j} = a_1(\|\hat{\mathbf{Y}}_{i,:}\|_2^2 - \frac{1}{c})(\|\hat{\mathbf{Y}}_{j,:}\|_2^2 - \frac{1}{c}) + a_2 \log(e - 1 + |\tilde{\mathbf{A}}_{i,j}|) + a_3 \text{Sigmoid}(a^T \text{ReLU}([h_i^{(0)} \| h_j^{(0)}])), \quad (13)$$

where $[a_1, a_2, a_3] = \text{Softmax}(\mathbf{W}_a), \mathbf{W}_a \in \mathbb{R}^{1 \times 3}$, $\|$ denotes vector concatenation, and $e$ is the natural logarithm.

The Euclidean norm term measures the quality of the pseudo label since a uniformly distributed $\|\hat{\mathbf{Y}}_{i,:}\|_2^2$ suggests that the estimator has no confidence in the class of $v_i$. The absolute value term takes into account that extreme values are typically more reliable. For example, a zero $\tilde{\mathbf{A}}_{i,j}$ indicates that the relationship between $v_i$ and $v_j$ is ambiguous. The third term employs graph attention [Velickovic et al., 2018, Brody et al., 2022] to adapt $c_{i,j}$ accordingly.

## 5 Experiment

This section evaluates the performance of LRGNN. We put the experimental results w.r.t. ablation study and robustness in the Supplement due to space limitation.

**Datasets.** We use three homophilous datasets including Cora, Citeseer and Pubmed [Yang et al., 2016], along with 6 heterophilous datasets released in Pei et al. [2020] and Rozemberczki et al. [2021], and three large-scale heterophilous graphs [Lim et al., 2021]. The training/validation/testing splits used in this paper are the same as [Li et al., 2022].

**Baselines.** We compare LRGNN with 11 baselines, including (1) classic GNN models: vanilla GCN [Kipf and Welling, 2017],and GAT [Velickovic et al., 2018]. (2) models specifically designed to handle heterophily: H$_2$GCN [Zhu et al., 2020], GPR-GNN [Chien et al., 2021], WRGAT [Suresh et al., 2021], LINKX [Lim et al., 2021], GGCN [Yan et al., 2021], ACM-GCN [Luan et al., 2021], GloGNN++ [Li et al., 2022], and Ordered GNN (OGNN) [Song et al., 2023]. (3) 2-layer MLP. We specifically choose GloGNN++ and ACM-GCN as they generally outperforme the other variants proposed in their respective papers.

**Node classification results.** Table 2 and Table 3 provide a summary of the test accuracy of the tested methods on 12 datasets with diverse homophily ratios and scales. Some baselines experience out-of-memory errors on large datasets, thus the corresponding results are excluded. From the tables, we can

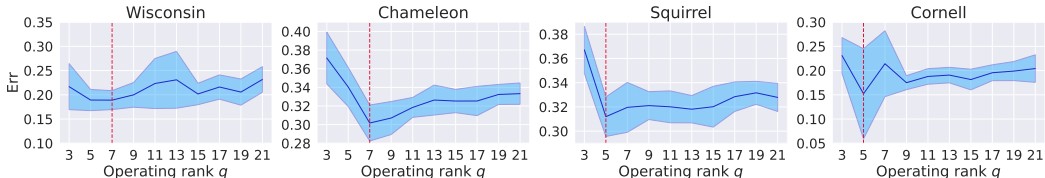

Figure 3: Recovery error. The lowest point is associated with a vertical line. The shaded region corresponds to a 95% confidence interval.

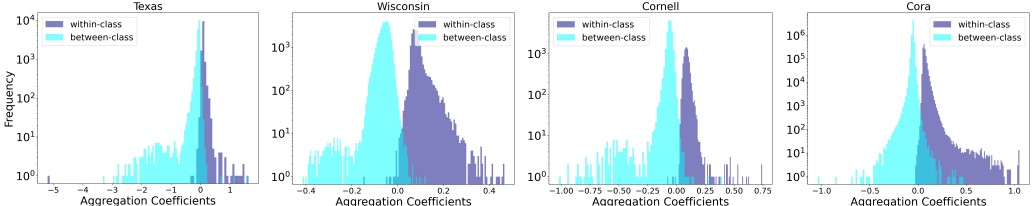

Figure 4: Visualization result of $\mathbf{Z}$. The entries have been grouped into two categories based on the label relationship between the two nodes.

draw several observations. Firstly, MLP is a strong baseline for heterophilous datasets, outperforming GCN and GAT on Texas, Wisconsin, and Cornell datasets. Secondly, methods designed specifically for heterophilous graphs generally perform better than MLP and traditional GNNs. Thirdly, $H_2$GCN, WRGAT, GGCN, ACM-GCN, GloGNN++, and LRGNN are the top performers on small datasets. However, $H_2$GCN, WRGAT, and GGCN lack scalability, thereby limiting their widespread adoption for large datasets. Fourthly, LRGNN performs the best in terms of the average rank, emerging as the winner or runner-up on all datasets except for Actor.

The results suggest that LRGNN consistently offers superior performance on both homophilous and heterophilous graphs. Notably, LRGNN achieved the highest score on Squirrel, with approximately 19% improvement over the runner-up score achieved by OGNN. The excellent performance on Squirrel and Chameleon could be attributed to the relatively large average node degrees of these two datasets, providing more observations to recover the label relationship matrix. Moreover, the performance of LRGNN on these three large datasets demonstrates the scalability of our model.

**Sign inference accuracy w.r.t. operating rank** $q$**.** We investigate the effect of the operating rank $q$ on the recovery loss: $err = \frac{1}{2n^2} \sum_{i,j} |sign((\mathbf{U}_* \mathbf{V}_*^T)_{i,j}) - sign(\hat{\mathbf{Z}}_{i,j})|$. 3 shows that the error decreases initially when $q$ increases, then it increases gradually after reaching the minimum point. The best result is consistently achieved when $q$ is approximately 5, the number of classes. Empirically, this supports the use of low-rank approximation to predict the label relationship matrix: the appropriate dimensionality for the factors can be determined based on the number of classes. In a standard LRMF problem, the rank of the matrix of interest is inaccessible. Therefore, we have to select a sufficiently large $q$ to ensure that $q$ surpasses the rank, which guarantees that the solution to LRMF problem provides a solution to the LRMC problem. Nevertheless, selecting excessively large $q$ can substantially increase the number of iterations required to converge.

**Visualizing the predicted label relationship matrix Z.** There are two main desirable traits of $\hat{\mathbf{Z}}$ we want $\mathbf{Z}$ to possess. The first is that within-class entries are positive while between-class entries are negative. This ensures that nodes are embedded close to their homophilous nodes and distant from the heterophilous nodes in the embedding space. The second is that $\hat{\mathbf{Z}}_{\mathbf{i},:}$ and $\hat{\mathbf{Z}}_{\mathbf{j},:}$ are identical if they share the same label, so that $\hat{\mathbf{Z}}_{\mathbf{i},:}\mathbf{X} = \hat{\mathbf{Z}}_{\mathbf{j},:}\mathbf{X}$. Consequently, we visualize $\mathbf{Z}$ in two different styles.

First, as shown in Figure 4, dividing the entries into two groups reveals that the distribution is similar to a Gaussian distribution. Although most entries converge around zero, within-class entries and between-class entries are positioned on opposite sides of the zero.

To illustrate the second trait, we visualize $\mathbf{Z}$ with reordered rows and columns based on ground truth labels. As shown in Figure 5, the matrices exhibit the structure of a block diagonal matrix. But here the entries of sub-matrices along the diagonal are positive, whereas off-diagonal sub-matrices contain

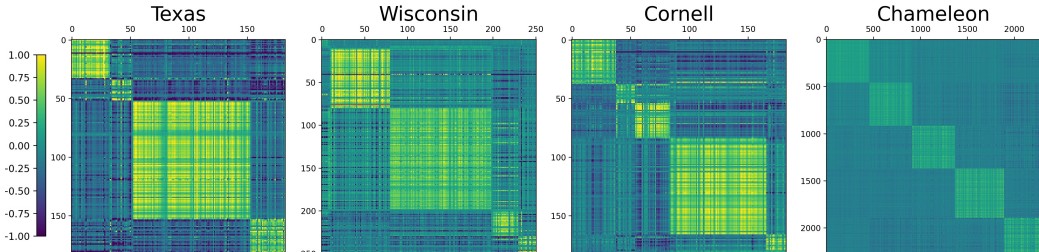

Figure 5: Visualization result of **Z**. Both row and column indices have been reordered based on the ground truth labels.

negative entries. On closer inspection of the figure, we notice that nodes in the same class share a high degree of similarity, with the rows' pixels showing comparable patterns. In conclusion, **Z** exhibits both desirable traits of **Ẑ**. This helps explain the effectiveness of LRGNN.

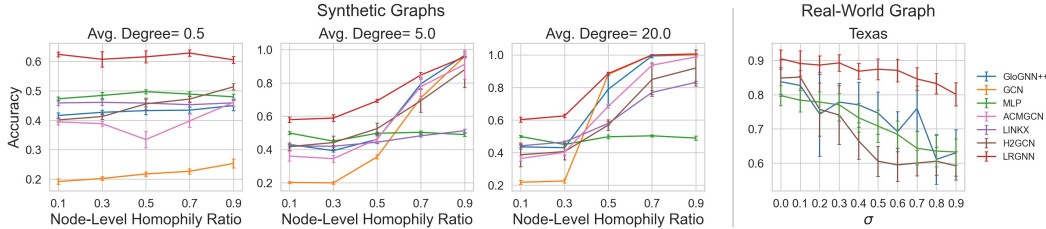

Figure 6: The first three subfigures are results on synthetic graphs and the last one is the result on corrupted Texas. Error bars indicate $95\%$ confidence interval.

**Results on synthetic graphs.** To comprehensively evaluate the performance of LRGNN, we leverage random partition graphs [Kim and Oh, 2021] generated by the stochastic block model. The node features are sampled from Gaussian distributions where the centers of clusters are vertices of a hypercube. Notably, the distance between Gaussian distribution means is small, thus, it is challenging to differentiate between node features of distinct classes. This is evident by MLP's unsatisfactory performance on the synthetic graphs. We use 15 synthetic random graphs with varying node-level homophily ratios and average degrees. More information about the synthetic graphs can be found in the Supplement.

It can be observed from Figure 6 that, when the homophily ratio or node degree is low, LRGNN is the only model that outperforms MLP. Also, LRGNN performs significantly better than other models when the homophily ratio is below $0.5$. These results suggest that LRGNN performs particularly well on random graphs, which we attribute to two reasons. Firstly, the observed entries of these graphs are uniformly distributed, meeting one of the conditions for exact recovery. Secondly, to derive aggregation coefficients, low-rank approximation methods utilize only the observed entries of **Ẑ**, while neural network based methods rely on the raw features. However, in this case, node features are not that informative since the means of Gaussian distributions are close. To validate the second reason, we use the Texas dataset and degrad the quality of its features by adding Gaussian variables. We obtain a degraded feature matrix by $\mathbf{X}'_{i,j} = \mathbf{X}_{i,j} + \epsilon_j$, with $\epsilon_j$ i.i.d. sampled from a Gaussian distribution $\mathcal{N}(0, \sigma^2)$. Note that the original features will be overwhelmed by the Gaussian random variables if a large $\sigma$ is applied. We compare the performance of representative models that included GloGNN++, H2GCN, and MLP with that of LRGNN. It can be observed that, as the features get less informative, the performance of GloGNN++, H2GCN, and MLP decreases significantly, while LRGNN's accuracy remains above $80\%$. In addition to that, a large $\sigma$ also makes their training process erratic, which is accompanied by significant error bars. These empirical results confirm our previous analysis.

# 6    Conclusion

This paper presents a method for extending GNNs to heterophilous graphs through the use of a label relationship matrix. In order to utilize the low-rank properties of weakly-balanced graphs, we propose a robust low-rank matrix approximation technique for the prediction of the label relationship matrix. The proposed LRGNN has been thoroughly evaluated through extensive experiments. We conduct extensive experiments to evaluate our proposed LRGNN. Our results indicate that LRGNN outperforms other baseline methods on both synthetic and real-world graphs. Notably, when the node degree distribution of the graph conforms to a uniform distribution, LRGNN exhibits significantly superior performance over other baseline methods.

# 7    Limitations

The main theoretical support for our approach assumes a uniform distribution of node degrees. However, in practice, node degree distributions frequently follow a power-law distribution that is non-uniform. Although the main results of this paper are obtained using real-world graphs with non-uniform node degree distributions, it is unclear how this affects the theoretical results. Additionally, it is acknowledged that LRGNN's performance is affected by the accuracy of the generated pseudo labels. The supplementary material provides the corresponding experiment that indicates LRGNN demonstrates a non-sensitive response to this impact. However, further exploration and discussion are helpful to investigate the theoretical implications fully.

## Acknowledgements

This work was partially supported by the National Key Research and Development Program of China (No. 2018AAA0100204), a key program of fundamental research from Shenzhen Science and Technology Innovation Commission (No. JCYJ20200109113403826), the Major Key Project of PCL (No. 2022ZD0115301), and an Open Research Project of Zhejiang Lab (NO.2022RC0AB04).

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
