# Appendices

## A  Low-Rank Matrix Factorization with Non-Uniform Sampling

In this section, we demonstrate the effectiveness of low-rank matrix factorization in recovering the label relationship matrix.

We first present four important facts:

f1: the rank of the matrix is equivalent to the number of classes.

f2 (homophilous node pair): if $\tilde{\mathbf{A}}_{i,j} = 1$, then $\hat{\mathbf{Z}}_{i,:} = \hat{\mathbf{Z}}_{j,:}$.

f3 (heterophilous node pair): if $\tilde{\mathbf{A}}_{i,j} = -1$, then $\hat{\mathbf{Z}}_{i,:} \neq \hat{\mathbf{Z}}_{j,:}$. Specifically, this also means that if $\hat{\mathbf{Z}}_{i,k} = 1$, then $\hat{\mathbf{Z}}_{j,k} = -1$.

f4 (symmetry): $\hat{\mathbf{Z}}_{i,j} = \hat{\mathbf{Z}}_{j,i}$.

We consider a toy example (without self-loops),

$$\hat{\mathbf{Z}} = \begin{bmatrix} 1 & -1 & -1 & 1 \\ -1 & 1 & -1 & -1 \\ -1 & -1 & 1 & -1 \\ 1 & -1 & -1 & 1 \end{bmatrix} \quad \tilde{\mathbf{A}} = \begin{bmatrix} 0 & -1 & -1 & 1 \\ 0 & 0 & -1 & 0 \\ 0 & 0 & 0 & 0 \\ 0 & 0 & 0 & 0 \end{bmatrix} \tag{14}$$

In a standard LRMF problem, it is not possible to recover $\hat{\mathbf{Z}}$ from $\tilde{\mathbf{A}}$ since no entries are observed for the third and fourth rows. However, we can demonstrate how LRMF effectively performs in this situation. Assuming we know that the number of classes is 3, we can obtain a solution $\mathbf{U}\mathbf{V}^T$, where each $\mathbf{U}_{i,:} \in \{1, -1\}^3$ is a 3D vector (f1), with $\mathbf{U}_{i,k} = 1$ if $v_i$ belongs to class $k$, and $\mathbf{V}_{i,:} \in \{0, 1\}^3$ is an indicator vector with $\mathbf{V}_{i,k} = 1$ indicating $v_i$ belonging to class $k$. This provides $\hat{\mathbf{Z}} = \mathbf{U}\mathbf{V}^T$.

**Recovery:** We begin by assuming $v_1$ is in class 1, resulting in $\mathbf{U}_{1,:} = [1, -1, -1]$ and $\mathbf{V}_{1,:} = [1, 0, 0]$. By observing $\tilde{\mathbf{A}}_{1,4}$, we know that $v_4$ is also in class 1, resulting in $\mathbf{U}_{4,:} = [1, -1, -1]$ and $\mathbf{V}_{4,:} = [1, 0, 0]$ (f2). By analyzing $\tilde{\mathbf{A}}_{1,2}$ and $\tilde{\mathbf{A}}_{1,3}$, we determine that $v_2$ and $v_3$ do not belong to class 1. We also know that since $\tilde{\mathbf{A}}_{2,3} = -1$, $v_2$ and $v_3$ must belong to different classes (f3). Therefore, we can assign

$$\mathbf{U}_{2,:} = [-1, 1, -1] \quad \mathbf{V}_{1,:} = [0, 1, 0]$$

and

$$\mathbf{U}_{3,:} = [-1, -1, 1] \quad \mathbf{V}_{1,:} = [0, 0, 1]$$

Put all together, we guess

$$\mathbf{U} = \begin{bmatrix} 1 & -1 & -1 \\ -1 & 1 & -1 \\ -1 & -1 & 1 \\ 1 & -1 & -1 \end{bmatrix} \quad \mathbf{V} = \begin{bmatrix} 1 & 0 & 0 \\ 0 & 1 & 0 \\ 0 & 0 & 1 \\ 1 & 0 & 0 \end{bmatrix}. \tag{15}$$

The correctness of this solution can be easily checked:

$$\hat{\mathbf{Z}} = \mathbf{U}\mathbf{V}^T \tag{16}$$

In this example, we successfully recover a $4 \times 4$ matrix despite having only 4 observed samples. Notably, the observations are not uniformly distributed, with 3 out of 4 located in the first row. Our approach involves using f1 to select $q = 3$ for $\mathbf{U}$ and $\mathbf{V}$, f2 to infer $\mathbf{U}_{4,:}$ and $\mathbf{V}_{4,:}$, and f3 to infer the second and third rows of $\mathbf{U}$ and $\mathbf{V}$. This simple illustration demonstrates that LRMF is particularly well-suited for label relationship matrix prediction tasks.

In the specific LRMF problems, we can teach the LRMF model these facts by adding these missing values to the observation set. Figure 7 shows that when the node degree distribution is non-uniform, the LRMF model only attains 69% accuracy. However, we achieve nearly perfect recovery from only

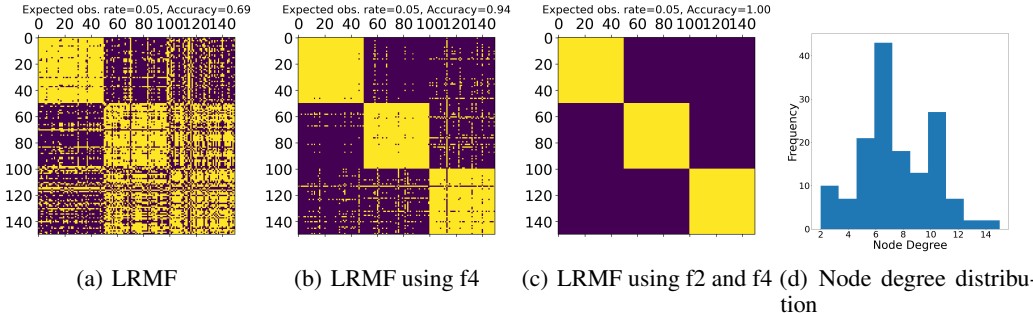

(a) LRMF  (b) LRMF using f4  (c) LRMF using f2 and f4  (d) Node degree distribution

Figure 7: LRMF results with an expected observation rate of 0.05.

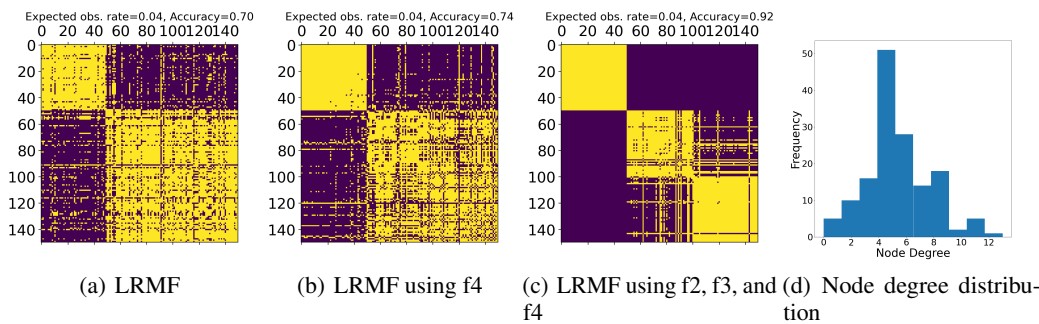

(a) LRMF  (b) LRMF using f4  (c) LRMF using f2, f3, and f4  (d) Node degree distribution

Figure 8: LRMF results with an expected observation rate of 0.05. The node degree distribution is highly skewed, making it similar to a power-law distribution.

5% observed entries by incorporating f4 into the LRMF model. Surprisingly, by further introducing f2, the label relationship matrix is predicted with $100\%$ accuracy. Figure 8 demonstrates that even with a more skewed node degree distribution and a lower observation rate, the model remains highly effective when using these facts. Exact recovery theoretically necessitates uniform sampling and a significant number of observations. In practice, it is possible to achieve perfect recovery of the label relationship matrix even with a limited number of observations and non-uniform sampling, thanks to the four previously mentioned facts.

Consequently, we conclude that LRMF serves as an excellent method for recovering the label relationship matrix, and the theoretical perfect recovery conditions are not always necessary. Our approach for integrating the four significant facts into the LRMF model allows perfect recovery from non-uniform samples.

## B  Impact of the Accuracy of the Estimated Signed Adjacency Matrix

This section analyzes the LRMF's sensitivity to the accuracy of the generated signed adjacency matrix. As previously demonstrated, we have established the effectiveness of the procedure for estimating the signed adjacency matrix. A precise estimation only requires moderately accurate pseudo-labels. This is because the focus lies on the label relationship between two nodes and not their precise classes. However, the objective of this section is to explore the model itself. LRGNN employs a specifically designed element-wise capped norm to identify incorrect estimations and assign smaller upper bounds to their contribution to the loss function.

In this section, an experiment is conducted to verify the effectiveness of the capped norm and to demonstrate the rationale for the design of $c_{i,j}$. The estimation error can be measured by $|y_{i,j} - \tilde{\mathbf{A}}_{i,j}|$, where $y_{i,j} = 1$ for matching labels and $-1$ otherwise. Ideally, a large $|y_{i,j} - \tilde{\mathbf{A}}_{i,j}|$ indicates that $\tilde{\mathbf{A}}_{i,j}$ is deviated from the ground truth label relationship $y_{i,j}$, thus it should be assigned a small $c_{i,j}$ value, improving the chance of detecting it as an outlier. In order to enable comparisons, we normalize $c_{i,j}$ using the formula $c_{i,j} = \frac{c_{i,j}}{\sum_{(i,j) \in \mathcal{E}} c_{i,j}}$. The capped norm is capable of identifying incorrect

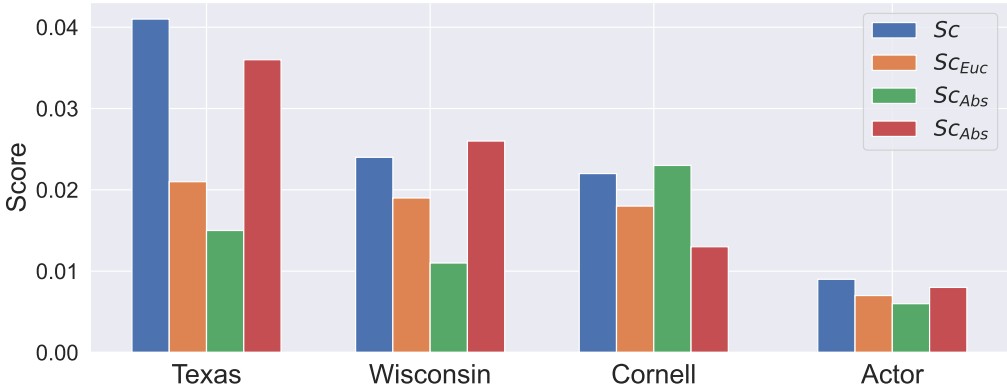

Figure 9: Gains achieved by the element-wise capped norm under different definitions of $\tau_{i,j}$.

estimations if and only if $Sc = \frac{1}{|\mathcal{E}|}\sum_{(i,j)\in\mathcal{E}}(1-c_{i,j})|y_{i,j} - \tilde{\mathbf{A}}_{i,j}| > 0$, provided that the average of $c_{i,j}$ is 1. To justify the design of the parameter $c_{i,j}$, we also execute an ablation study on the three components of $c_{i,j}$. We denote the removal of the Euclidean term from $c_{i,j}$ as $Sc_{Euc}$. Likewise, $Sc_{Abs}$ and $Sc_{Att}$ represent the removal of the absolute value term and attention term, respectively. Figure 9 demonstrates the results. We observe a consistent gain ($S_c > 0$) for all datasets resulting from the use of the capped norm. The findings suggest that our outlier detection algorithm can identify untrustworthy estimations and assign more restrictive thresholds to them. Consequently, LRGNN can effectively mitigate the impact of inaccurate estimations and limit their effect on empirical loss. The ablation study also provides evidence for the three components used in $c_{i,j}$.

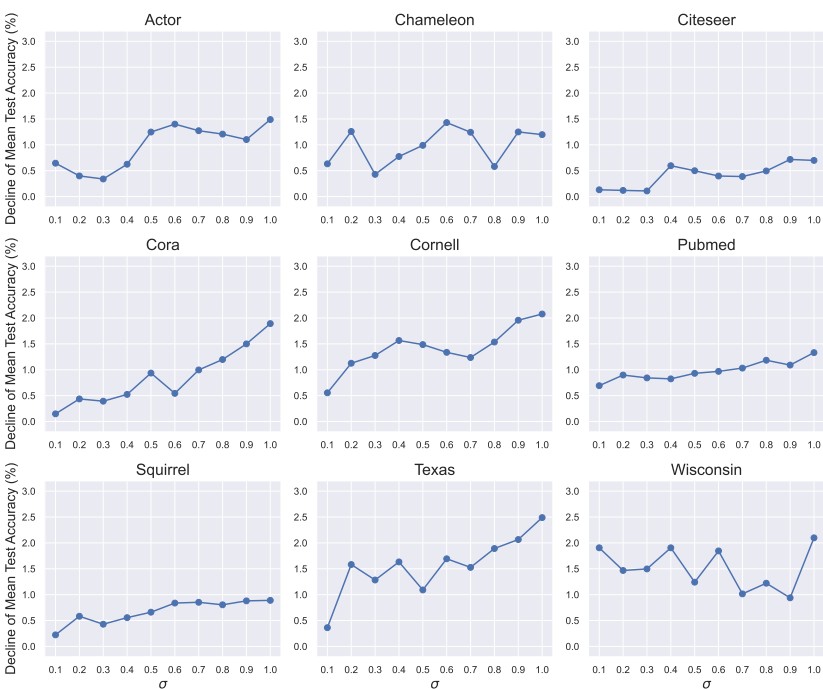

Figure 10: The mean test accuracy decreases as the standard deviation of the Gaussian noise increases.

To empirically validate the robustness, we conduct further analysis of LRGNN's response to a low-quality signed adjacency matrix by introducing random noise. This reproduces a scenario in which the signed adjacency matrix is randomly distorted by Gaussian noise. To be precise, we create a noisy

signed adjacency matrix, $\mathbf{A}_{noise}$, from the original matrix $\tilde{\mathbf{A}}$ by appending a Gaussian noise matrix $\mathbf{N}$. In $\mathbf{N}$, we set the value of $\mathbf{N}_{i,j}$ to $\epsilon_{i,j}$, if $(i,j) \in \mathcal{E}$ and zero otherwise, where $\epsilon_{i,j}$ is independently and identically sampled from a $\mathcal{N}(0, \sigma^2)$ distribution. We select $\sigma$ from the set $\{0.1, 0.2, ..., 1.0\}$. We present findings on the mean test accuracy reduction of LRGNN between using a corrupted signed adjacency matrix and using a normal one in Figure 10. We observe that for all datasets, the declines are less than $3\%$, regardless of the value of $\sigma$. It is important to note that entries of $\tilde{\mathbf{A}}$ fall within the range of -1 to 1, and the noise may cause significant changes when $\frac{|\epsilon_{i,j}|}{|\tilde{\mathbf{A}}_{i,j}|} > 1$. The results suggest that the quality of the generated signed adjacency matrix has a not critical impact on LRGNN's performance, and the outstanding performance of LRGNN is not conditioned on a very accurate signed adjacency matrix. This can be credited to the element-wise capped norm.

# C Additional Experimental Results

## C.1 Spectral Clustering with Z

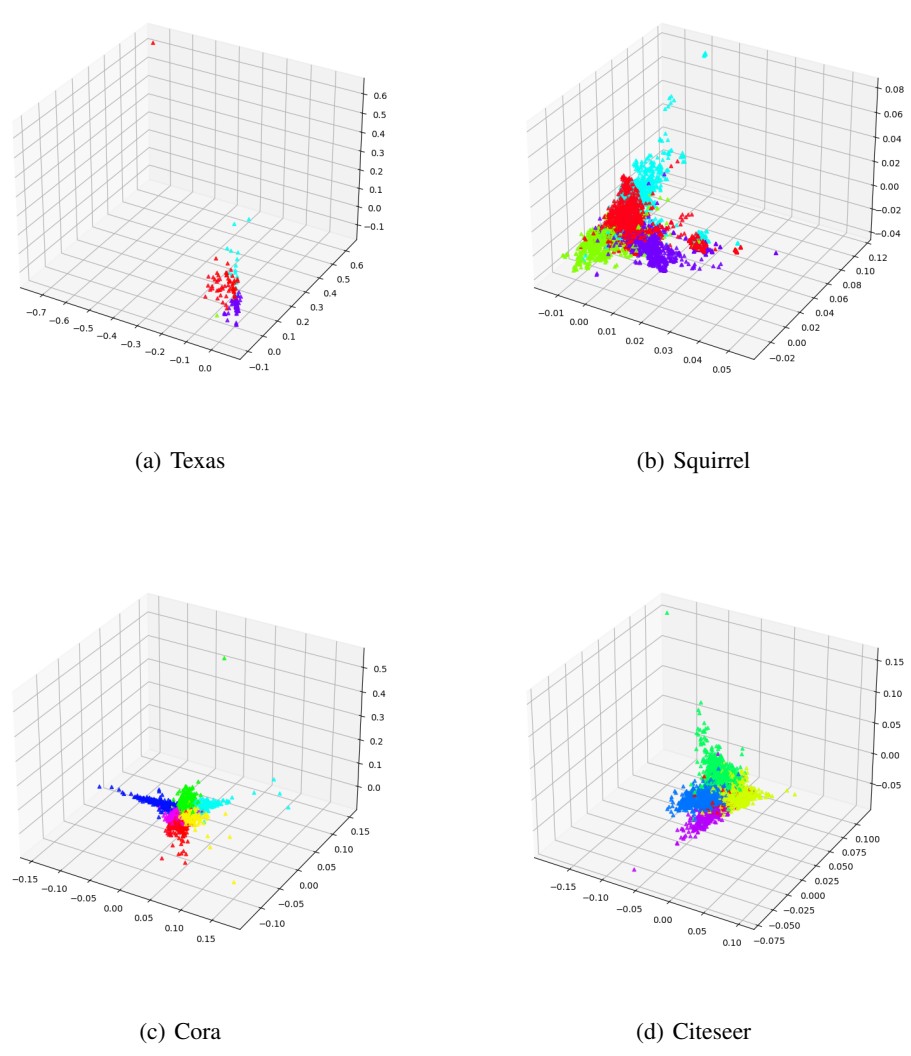

(a) Texas

(b) Squirrel

(c) Cora

(d) Citeseer

Figure 11: Visualization results of the first three eigenvectors by rows.

This section demonstrates how to take advantage of $\mathbf{Z}$ to identify clusters. We show that the true clusters can be identified from the eigenvectors of $\mathbf{Z}$. This is because the eigenvectors of $\hat{\mathbf{Z}}$ have a nice property. Let $\mathbf{u} \in \mathbb{R}^{n \times 1}$ be an arbitrary eigenvector of $\hat{\mathbf{Z}}$ with an associated eigenvalue $\lambda$ being nonzero. $\mathbf{u}$ can reveal the membership of each node as $u_i = u_j$ iff $v_i$ and $v_j$ are in the same cluster. Suppose $v_i$ and $v_j$ are in the same cluster, we have $\hat{\mathbf{Z}}_{i,:}\mathbf{u} = \lambda u_i$ and $\hat{\mathbf{Z}}_{j,:}\mathbf{u} = \lambda u_j$. Since $\hat{\mathbf{Z}}_{i,:} = \hat{\mathbf{Z}}_{j,:}$, this leads to $u_i = u_j$. On the other hand, if $v_i$ and $v_j$ belong to different clusters, i.e., $\hat{\mathbf{Z}}_{i,:} \neq \hat{\mathbf{Z}}_{j,:}$, then $u_i \neq u_j$. To illustrate this property, we visualize the eigenvectors associated with the largest three eigenvalues of $\mathbf{Z}$ obtained from different datasets. specifically, we construct a matrix $\in \mathbb{R}^{n \times 3}$ with these three eigenvectors as columns and utilize each node's corresponding 3D row vector as its coordinate. Figure 11 shows the visualization results. The nodes in the same cluster are agglomerated together, while the nodes of different clusters are split apart, demonstrating a compelling clustering effect. This empirically confirms LRMF's effectiveness in spectral clustering, thus enabling us to employ the derived $\mathbf{Z}$ for graph spectral clustering.

## C.2 Impact of Parameter $\lambda$ on the Rank of the Solution

In this part, we investigate the impact of the regularization parameter $\lambda$. First, we will demonstrate the connection between LRMF and the nuclear-norm regularized problem. Mathematically, the LRMF problem

$$\min_{\mathbf{U}_{m \times r}, \mathbf{V}_{n \times r}} \quad \frac{1}{2}\|\mathbf{P}_\Omega(\mathbf{X} - \mathbf{U}\mathbf{V}^T)\|_F^2 + \frac{\lambda}{2}(\|\mathbf{U}\|_F^2 + \|\mathbf{V}\|_F^2), \tag{17}$$

and the nuclear-norm regularized problem

$$\min_{\mathbf{Z}_{m \times n}} \quad \frac{1}{2}\|\mathbf{P}_\Omega(\mathbf{X} - \mathbf{Z})\|_F^2 + \lambda\|\mathbf{Z}\|_*, \tag{18}$$

can be connected by the following theorem.

*Theorem* 3. [Mazumder et al., 2010] Let $\mathbf{X}$ be a $m \times n$ matrix with observed entries indexed by $\Omega$. Let $r = \min(m, n)$, then the solutions to (17) and (18) coincide for all $\lambda \geq 0$.

This is the direct result of the following lemma (see, e.g., Srebro et al. [2004] for proof).

$$\|\mathbf{X}\|_* = \min_{\mathbf{X} = \mathbf{U}\mathbf{V}^T} \quad \frac{1}{2}(\|\mathbf{U}\|_F^2 + \|\mathbf{V}\|_F^2). \tag{19}$$

The above lemma implies that one can bound the trace norm of $\mathbf{U}\mathbf{V}^T$ using Frobeniums norm regularization $\frac{1}{2}(\|\mathbf{U}\|_F^2 + \|\mathbf{V}\|_F^2)$. Thus, the $\lambda$ parameter of the LRMF problem serves as a regularization parameter responsible for controlling the nuclear norm of the solution. Raising $\lambda$ will eventually decrease the rank of the solution. To substantiate this, we have illustrated the ranks of the matrices $\mathbf{U}$, $\mathbf{V}$, $\mathbf{Z}$, and $\hat{\mathbf{Z}}$ in Figure 12, concerning differing values of $\lambda$. It is clear from the figure that the ranks of these matrices diminish considerably as $\lambda$ increases. Furthermore, it is worthwhile to note that when $\lambda$ is small, the ranks of $\mathbf{U}$ and $\mathbf{V}$ differ significantly from the true rank of 3. However, the rank of $\mathbf{Z}$ is much smaller than that of $\mathbf{U}$ and $\mathbf{V}$. Notably, when $\lambda$ is set to 5, 10, and 20, $\mathbf{Z}$ and $\hat{\mathbf{Z}}$ exhibit equal ranks, which means that LRMF provides a good solution. Figure 13 illustrates that the change in training and test MSE loss follows a similar pattern as the changes in ranks. The optimal result is achieved when the rank of $\mathbf{Z}$ is the same as that of $\hat{\mathbf{Z}}$.

## C.3 Ablation Study.

In this section, we perform an ablation analysis to evaluate the effectiveness of each component in the LRGNN model. Specifically, we examine five variants: (1) LRGNN-MF, which excludes the attention term from the objective function, (2) LRGNN-Uni, which replaces the signed adjacency matrix with the uniform sparse adjacency matrix used in GCN, (3) LRGNN-Reg, which replaces the matrix factorization term with a regularization term and removes the projection function, (4) LRGNN-DA, which exclusively relies on node features to generate the initial node representations $\mathbf{H}^{(0)}$, and (5) LRGNN-NC, which drops the capped norm from the objective function. We compare the performance of these variants with that of LRGNN and report the node classification results in Table 4. Our analysis reveals that LRGNN-Reg and LRGNN-Uni are the least effective of the five variants. This observation aligns with our intuition, as LRGNN-Reg and LRGNN-Uni impair the LRMF model, while other components are the complements to LRMF model. Therefore, removing them has no direct impact on the LRMF model.

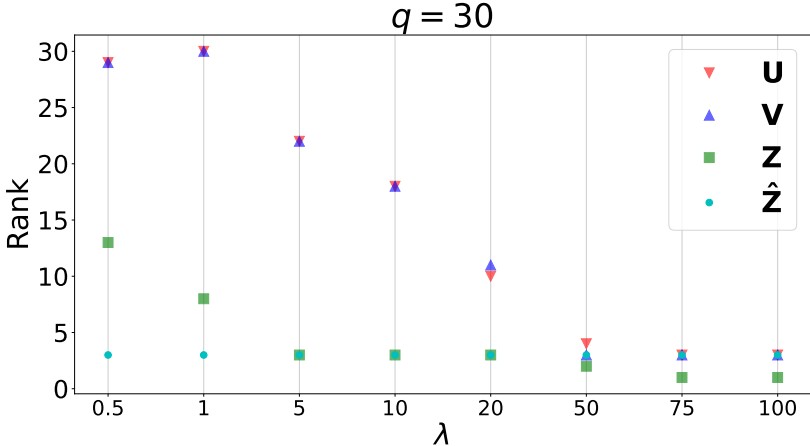

Figure 12: The rank of the matrices obtained when different $\lambda$ are applied.

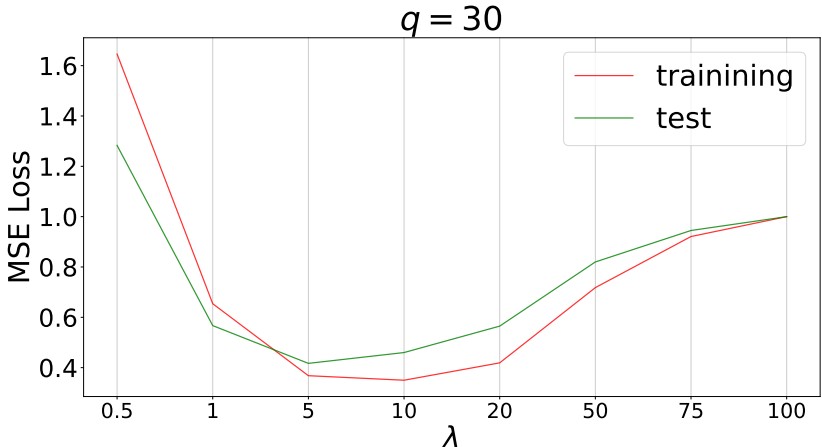

Figure 13: Training and test mean square error losses with respect to different $\lambda$ values.

### C.4   More Visualization Results

In this section, we decompose the derived matrix $\mathbf{Z}$ into two parts: $\mathbf{Z} = \text{MF} + \text{Att}$, where

$$\text{MF} = (\widehat{\mathbf{A}}\mathbf{V}^{(k)})[(1+\gamma)\mathbf{V}^{(k)^T}\mathbf{V}^{(k)} + \lambda\mathbf{I}_q]^{-1}\mathbf{V}^{(k+1)^T}, \qquad (20)$$

$$\text{Att} = (\gamma\mathbf{H}^{(0)}\mathbf{W}\mathbf{H}^{(0)^T}\mathbf{V}^{(k)})[(1+\gamma)\mathbf{V}^{(k)^T}\mathbf{V}^{(k)} + \lambda\mathbf{I}_q]^{-1}\mathbf{V}^{(k+1)^T}, \qquad (21)$$

such that $\mathbf{Z} = \text{MF} + \text{Att}$, where MF denotes the matrix factorization term while Att denotes the attention term. We analyze the contributions of two matrices to enhance our understanding of their respective roles. Figure 14 reveals that the MF matrix exhibits a prominent block-diagonal

| | Texas | Wisconsin | Cornell | Actor | Squirrel | Chamel. | Cora | Citeseer | Pubmed |
|---|---|---|---|---|---|---|---|---|---|
| LRGNN | **90.27±4.49** | **88.23±3.54** | **86.22±6.50** | **37.34±1.78** | **74.38±1.96** | **79.16±2.05** | **88.33±0.89** | **77.53±1.31** | **90.16±0.64** |
| LRGNN-MF | 89.19±4.90 | 84.71±4.75 | 83.51±6.32 | 35.69±0.95 | 68.23±4.38 | 71.48±4.06 | 87.88±1.03 | 77.30±1.39 | 89.05±0.43 |
| LRGNN-Reg | 88.38±2.43 | 82.94±4.11 | 83.51±5.85 | 36.61±1.17 | 65.83±2.19 | 69.12±0.84 | 86.92±0.96 | 75.57±1.60 | 87.36±0.25 |
| LRGNN-Uni | 87.84±3.67 | 82.94±4.72 | 81.89±5.13 | 34.53±1.08 | 69.45±1.78 | 68.46±1.38 | 87.67±1.38 | 77.29±1.29 | 88.62±0.18 |
| LRGNN-DA | 89.19±4.15 | 86.86±3.58 | 85.95±7.66 | 36.86±1.08 | 72.52±1.78 | 75.65±1.38 | 88.23±1.38 | 77.32±1.29 | 89.45±0.18 |
| LRGNN-NC | 88.34±2.89 | 86.86±3.89 | 85.14±6.53 | 36.10±1.48 | 73.12±2.06 | 77.38±1.68 | 87.87±1.68 | 76.32±1.02 | 89.06±0.25 |

Table 4: Ablation study.

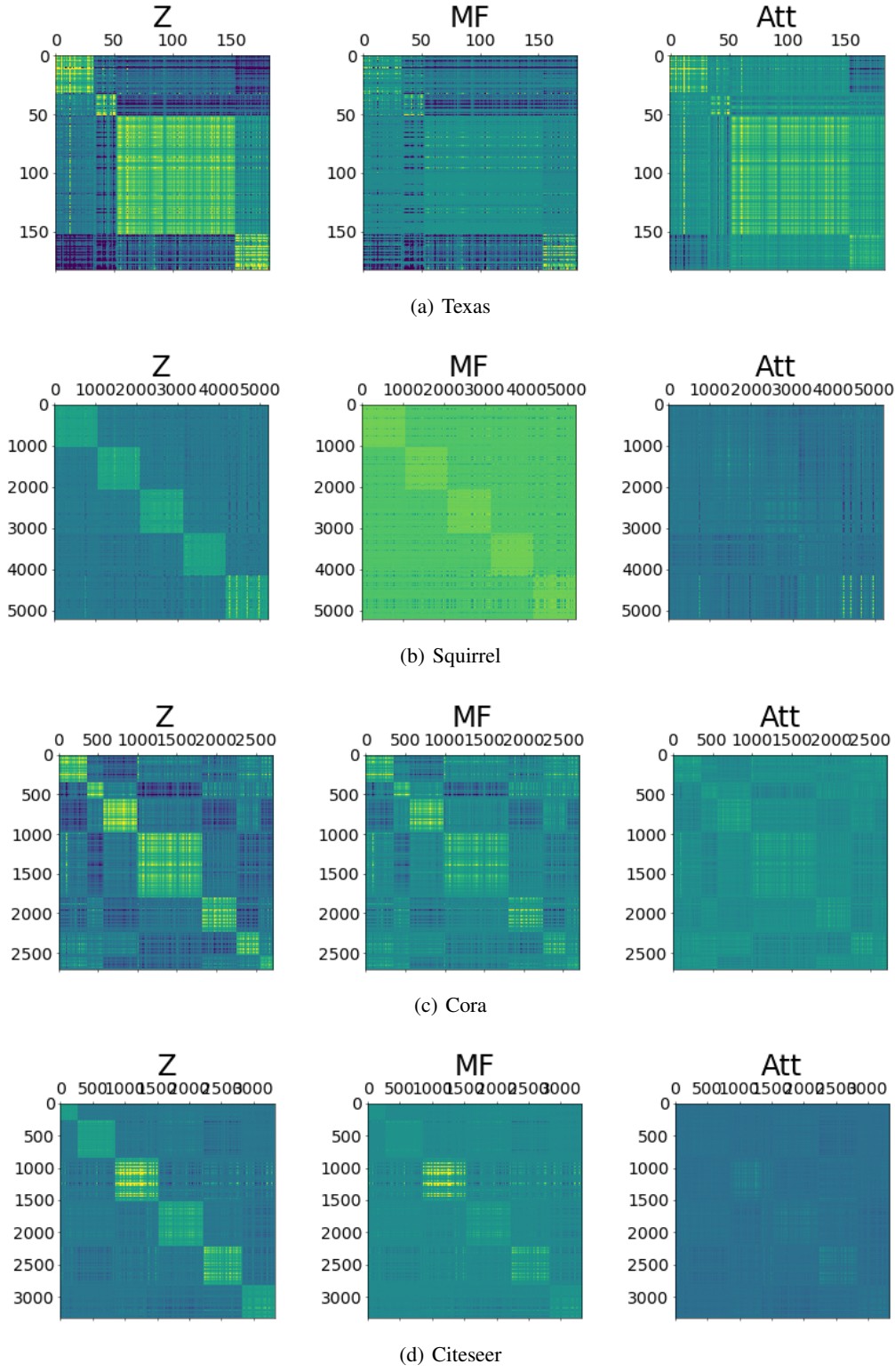

(a) Texas

(b) Squirrel

(c) Cora

(d) Citeseer

Figure 14: Decomposing $\mathbf{Z}$ into matrix factorization (MF) part and attention part (Att) as $\mathbf{Z} = \text{MF} + \text{Att}$.

|  | roman-empire | amazon-ratings | minesweeper | tolokers | questions |
|---|---|---|---|---|---|
| GCN | 73.69 | 48.70 | 89.75 | 83.64 | 76.09 |
| GAT | 80.87 | 49.09 | 92.01 | **83.70** | 77.43 |
| SAGE | **85.74** | **53.63** | **93.51** | 82.43 | 76.44 |
| H2GCN | 60.11 | 36.47 | 89.71 | 73.35 | 63.59 |
| GPR-GNN | 64.85 | 44.88 | 86.24 | 72.94 | 55.48 |
| FSGNN | 79.92 | 52.74 | 90.08 | 82.76 | 78.86 |
| GloGNN | 59.63 | 36.89 | 51.08 | 73.39 | 65.74 |
| FAGCN | 65.22 | 44.12 | 88.17 | 77.75 | 77.24 |
| LRGNN | 81.27 | 49.43 | 91.62 | 75.72 | **79.53** |

Table 5: Node classification accuracy (%).

structure, while the structure of the Att matrix appears ambiguous. This indicates that the efficiency of LRGNN primarily originates from the MF term of the objective function. Nevertheless, an improved performance due to the Att matrix is also evident. In the case of the Squirrel dataset, the bright entries of the MF matrix predominantly express homophilous information. Conversely, the dark entries of the Att matrix convey the heterophilous information. This combination of the MF and Att matrices allows the model to leverage both homophilous and heterophilous information, leading to an overall improvement in expressiveness. Additionally, for all datasets, $\mathbf{Z}$ demonstrates superior structural coherence compared to the MF and Att matrices, judged by both its block-diagonal appearance and the distribution patterns of within- and between- class entries therein. In summary, while the MF term plays a crucial part in powering LRGNN, the attention term also strengthens the model.

### C.5 Results from Additional Datasets

A recent study [Platonov et al., 2023] highlights the limitations of commonly used datasets in evaluating GNNs under heterophily, such as small scale and the potential leakage of testing data. They also propose new datasets to address these issues and advance evaluation practices. Here we present node classification results obtained from these newly introduced datasets. As indicated in Platonov et al. [2023], traditional GNNs including GCN, GAT, and SAGE generally perform better than heterophily-specific models on these datasets. Our results in Table 5 show that the performance of LRGNN is comparable to FSGNN and surpasses other heterophily-specific models.

## D Optimization Algorithm

### D.1 An Iterative Method

Practical solutions to LRMF fall into two main camps, *i.e.*, Alternating Least Squares algorithm (ALS) [Wiberg, 1976, Shum et al., 1995, Huynh et al., 2003] and Newton methods [Buchanan and Fitzgibbon, 2005, Okatani and Deguchi, 2007, Chen, 2008]. The basic idea of ALS is that once one of $\mathbf{U}$ and $\mathbf{V}$ is fixed, (5) is convex in the other one. ALS decouples this problem into n separate ridge regressions. Since the positions of the missing values of each column are different, ALS needs to separately solve these n ridge regressions. Albeit Newton methods can obtain global optimum within each dimension subset at each descent step, they take a long time per iteration. We refer the interested reader to Davenport and Romberg [2016] for a survey. Although these methods can provide accurate recovery in general, they need to compute SVD on a $n \times n$ coefficient matrix. We adopt a SVD-free algorithm called SoftImpute-ALS [Hastie et al., 2015].

Most traditional approaches randomly initialize $\mathbf{U}$ and $\mathbf{V}$. In this paper, we employ a more neural-style initialization manner.

$$\mathbf{U}^{(0)} = \underset{\mathbf{U}}{argmin} \|\tilde{\mathbf{A}} - \mathbf{U}\mathbf{V}^{(0)}\|_2^2, \quad \mathbf{V}^{(0)} = f_{init}(\mathbf{H}^{(0)}) \qquad (22)$$

Here $f_{init}(\cdot)$ denotes a fully-connected layer or graph convolution layer, depending on the homophily ratio. We empirically found that this initialization can provide better results within a few iterations for updates.

Inspired by the reweighted methods Nie et al. [2014, 2017], we first construct a simple surrogate function for optimizing Eq.(8)

$$F_s(\mathbf{U}, \mathbf{V}) = \sum_{(i,j)\in\mathcal{E}} s_{i,j}((\mathbf{U}\mathbf{V}^T)_{i,j} - \tilde{\mathbf{A}}_{i,j})^2 + \lambda\|\mathbf{U}\|_F^2 + \lambda\|\mathbf{V}\|_F^2 + \gamma\sum_{i,j}(h_i^{(0)^T}\mathbf{W}^{(l)}h_j^{(0)} - (\mathbf{U}\mathbf{V}^T)_{i,j})^2,$$

$$(23)$$

where $s_{i,j} = 1$ if $e_{i,j} = ((\mathbf{U}\mathbf{V}^T)_{i,j} - \tilde{\mathbf{A}}_{i,j})^2 < \tau_{i,j}$ and $s_{i,j} = 0$ otherwise. Consider that we have current estimates $\mathbf{U}^{(k)}$ and $\mathbf{V}^{(k)}$, and wish to derive a new $\mathbf{U}^{(k+1)}$ that minimizes the objective function. Specifically, we introduce the following surrogate function for deriving $\mathbf{U}^{(k+1)}$.

$$\mathbf{S}_U(\mathbf{Z}_U|\mathbf{U}^{(k)}, \mathbf{V}^{(k)}) = \|\mathbf{M} \odot (\mathbf{Z}_U\mathbf{V}^{(k)^T} - \tilde{\mathbf{A}})\|_F^2 +$$

$$\|(\mathbf{1} - \mathbf{M}) \odot (\mathbf{Z}_U\mathbf{V}^{(k)^T} - \mathbf{U}^{(k)}\mathbf{V}^{(k)^T})\|_F^2 + \lambda\|\mathbf{Z}_U\|_F^2 + \lambda\|\mathbf{V}^{(k)}\|_F^2 + \gamma\|\mathbf{H}^{(0)}\mathbf{W}\mathbf{H}^{(0)^T} - \mathbf{Z}_U\mathbf{V}^{(k)^T}\|_F^2,$$

$$(24)$$

where $\mathbf{M}$ is defined as $\mathbf{M}_{i,j} = s_{i,j}$ if $(i, j) \in \mathcal{E}$ and $\mathbf{M}_{i,j} = 0$ otherwise, and $\odot$ refers to Hadamard product. Then $\mathbf{U}^{(k+1)}$ can be obtained by minimizing this surrogate function over $\mathbf{Z}_U$, *i.e.*, $\mathbf{U}^{(k+1)} = \underset{\mathbf{Z}_U\in\mathbb{R}^{n\times q}}{argmin} \ \mathbf{S}_U(\mathbf{Z}_U|\mathbf{U}^{(k)}, \mathbf{V}^{(k)})$. Note that $\|\mathbf{M} \odot (\mathbf{Z}_U\mathbf{V}^{(k)^T} - \tilde{\mathbf{A}})\|_F^2 + \|(\mathbf{1} - \mathbf{M}) \odot (\mathbf{Z}_U\mathbf{V}^{(k)^T} - \mathbf{U}^{(k)}\mathbf{V}^{(k)^T})\|_F^2 = \|\mathbf{Z}_U\mathbf{V}^{(k)^T} - (\mathbf{M} \odot \tilde{\mathbf{A}} + (\mathbf{1} - \mathbf{M}) \odot (\mathbf{U}^{(k)}\mathbf{V}^{(k)^T}))\|_F^2$, wherein $\mathbf{Z}_U$ gets rid of the Hadamard product, thus we can directly obtain the closed-form solution

$$\mathbf{U}^{(k+1)} = [\widehat{\mathbf{A}}\mathbf{V}^{(k)} + \gamma\mathbf{H}^{(0)}\mathbf{W}\mathbf{H}^{(0)^T}\mathbf{V}^{(k)}] \cdot [(1+\gamma)\mathbf{V}^{(k)^T}\mathbf{V}^{(k)} + \lambda\mathbf{I}_q]^{-1}, \quad (25)$$

where $\widehat{\mathbf{A}} = \mathbf{M} \odot \tilde{\mathbf{A}} + (\mathbf{1} - \mathbf{M}) \odot (\mathbf{U}^{(k)}\mathbf{V}^{(k)^T})$. Given $\mathbf{U}^{(k+1)}$, we define the following surrogate function for deriving $\mathbf{V}^{(k+1)}$,

$$\mathbf{S}_V(\mathbf{Z}_V|\mathbf{U}^{(k+1)}, \mathbf{V}^{(k)}) = \|\mathbf{M} \odot (\mathbf{U}^{(k+1)}\mathbf{Z}_V^T - \tilde{\mathbf{A}})\|_F^2 + \|(\mathbf{1} - \mathbf{M}) \odot (\mathbf{U}^{(k+1)}\mathbf{Z}_V^T - \mathbf{U}^{(k+1)}\mathbf{V}^{(k)^T})\|_F^2 +$$

$$\lambda\|\mathbf{Z}_V\|_F^2 + \lambda\|\mathbf{U}^{(k+1)}\|_F^2 + \gamma\|\mathbf{H}^{(0)}\mathbf{W}\mathbf{H}^{(0)^T} - \mathbf{U}^{(k+1)}\mathbf{Z}_V^T\|_F^2, \quad (26)$$

Similarly, $\mathbf{V}^{(k+1)}$ can be obtained by minimizing the above surrogate function over $\mathbf{Z}_V$. The closed-form solution is

$$\mathbf{V}^{(k+1)} = [\widehat{\mathbf{A}}^T\mathbf{U}^{(k+1)} + \gamma\mathbf{H}^{(0)}\mathbf{W}^T\mathbf{H}^{(0)^T}\mathbf{U}^{(k+1)}] \cdot [(1+\gamma)\mathbf{U}^{(k+1)^T}\mathbf{U}^{(k+1)} + \lambda\mathbf{I}_q]^{-1}. \quad (27)$$

Here $\widehat{\mathbf{A}} = \mathbf{M} \odot \tilde{\mathbf{A}} + (\mathbf{1} - \mathbf{M}) \odot (\mathbf{U}^{(k+1)}\mathbf{V}^{(k)^T})$. A main advantage of this optimization algorithm is that $\widehat{\mathbf{A}}$ can be decomposed as *sparse* plus *low rank* structure, which greatly accelerates the computation. Figure 15 depicts the empirical convergence rate. We define the change in Frobenius norm as $\frac{\|\mathbf{U}^k - \mathbf{U}^{k-1}\|_F^2}{\|\mathbf{U}^k\|_F^2}$, where $k$ is the $k$-th iteration. The results reveal that matrices $\mathbf{U}$ and $\mathbf{V}$ both almost converge after eight iterations.

### D.2  Time Complexity

Since all the involved matrices are low-rank or sparse, we can reduce the computation time using some tricks. Although $\widehat{\mathbf{A}}$ is not sparse, $\widehat{\mathbf{A}}\mathbf{V}^{(k)}$ can be decomposed into sparse-dense matrix multiplications, resulting in time complexity of $O(mq + nq^2)$, where $m$ denotes the number of edges.

We first consider the cost of updating $\mathbf{U}^{(k+1)}$.

$$\mathbf{U}^{(k+1)} = [\widehat{\mathbf{A}}\mathbf{V}^{(k)} + \gamma\mathbf{H}^{(0)}\mathbf{W}\mathbf{H}^{(0)^T}\mathbf{V}^{(k)}] \cdot [(1+\gamma)\mathbf{V}^{(k)^T}\mathbf{V}^{(k)} + \lambda\mathbf{I}_q]^{-1} \quad (30)$$

$\mathbf{V}^{(k)}$ and $\mathbf{H}^{(0)}$ are matrices of size $n \times q$ and size $n \times c$, respectively. For low-rank matrices, we can reorder the matrix multiplication. For example, the time complexity of the following computing order is $O(mq + q^3 + ncq + nq^2)$

$$\mathbf{U}^{(k+1)} = [\widehat{\mathbf{A}}\mathbf{V}^{(k)} + \gamma(\mathbf{H}^{(0)}\mathbf{W})(\mathbf{H}^{(0)^T}\mathbf{V}^{(k)})] \cdot [(1+\gamma)\mathbf{V}^{(k)^T}\mathbf{V}^{(k)} + \lambda\mathbf{I}_q]^{-1} \quad (31)$$

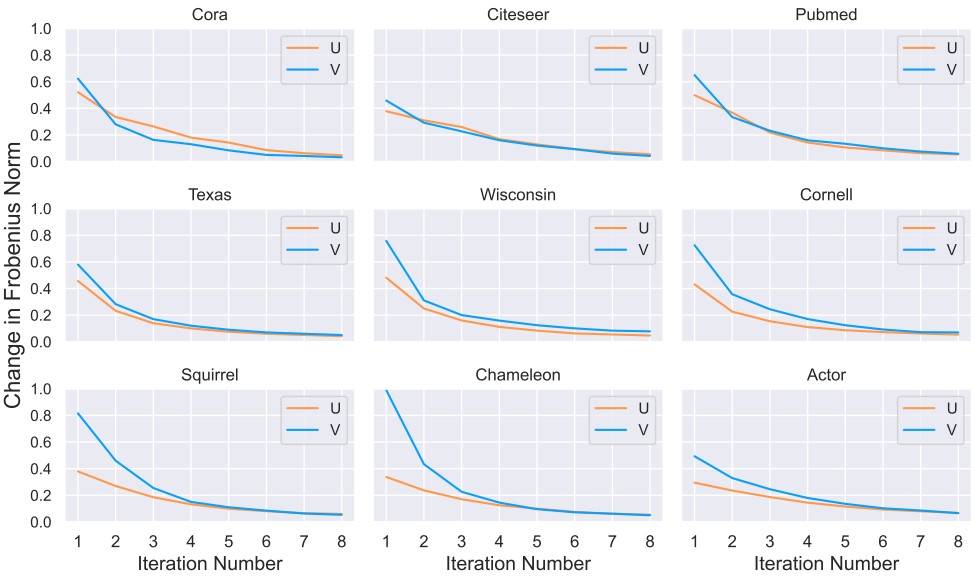

Figure 15: Convergence rate of the optimization algorithm.

---

**Algorithm 1** Algorithm to minimize Eq. (8)

---

1: Initialize $\mathbf{V}^{(0)}$ and $\mathbf{U}^{(0)}$ using Eq. (22)
2: **for** $k = 0$ to $K - 1$ **do**
3:     For $(i, j) \in \mathcal{E}$, reweight by calculating

$$s_{i,j} = \begin{cases} 1, (\mathbf{U}^{(k)}\mathbf{V}^{(k)^T} - \tilde{\mathbf{A}}_{i,j})^2 < \tau_{i,j} \\ 0, \quad \text{otherwise.} \end{cases} \quad (28)$$

4:     Compute $\mathbf{U}^{(k+1)}$ using Eq. (25)
5:     For $(i, j) \in \mathcal{E}$, reweight by calculating

$$s_{i,j} = \begin{cases} 1, (\mathbf{U}^{(k+1)}\mathbf{V}^{(k)^T} - \tilde{\mathbf{A}}_{i,j})^2 < \tau_{i,j} \\ 0, \quad \text{otherwise.} \end{cases} \quad (29)$$

6:     Compute $\mathbf{V}^{(k+1)}$ using Eq. (27)
7: **end for**
8: **return** $\mathbf{U}^{(K)}$ and $\mathbf{V}^{(K)}$.

---

where $\widehat{\mathbf{A}}\mathbf{V}^{(k)}$ can be realized by sparse-dense matrix multiplications.

$$\widehat{\mathbf{A}}\mathbf{V}^{(k)} = (\mathbf{M} \odot \tilde{\mathbf{A}} + (1 - \mathbf{M}) \odot (\mathbf{U}^{(k)}\mathbf{V}^{(k)^T}))\mathbf{V}^{(k)} = (\mathbf{M} \odot \tilde{\mathbf{A}} + \mathbf{U}^{(k)}\mathbf{V}^{(k)^T} - \mathbf{M} \odot (\mathbf{U}^{(k)}\mathbf{V}^{(k)^T}))\mathbf{V}^{(k)} \quad (32)$$

where $(\mathbf{M} \odot \tilde{\mathbf{A}})\mathbf{V}^{(k)}$ and $(\mathbf{M} \odot (\mathbf{U}^{(k)}\mathbf{V}^{(k)^T}))\mathbf{V}^{(k)}$ are sparse-dense matrix multiplications. the time complexity of $\mathbf{U}^{(k)}(\mathbf{V}^{(k)^T}\mathbf{V}^{(k)})$ is $O(nq^2)$.

Now we consider the calculating of $\mathbf{V}^{(k+1)}$

$$\mathbf{V}^{(k+1)} = [\widehat{\mathbf{A}}^T\mathbf{U}^{(k+1)} + \gamma\mathbf{H}^{(0)}\mathbf{W}^T\mathbf{H}^{(0)^T}\mathbf{U}^{(k+1)}] \cdot [(1 + \gamma)\mathbf{U}^{(k+1)^T}\mathbf{U}^{(k+1)} + \lambda\mathbf{I}_q]^{-1}. \quad (33)$$

The time complexity of the above equation is also $O(mq + q^3 + ncq + nq^2)$. Since the size of $\mathbf{V}^{(k+1)^T}\mathbf{H}^{(l)}$ is $q \times c$, $\mathbf{U}^{(k+1)}(\mathbf{V}^{(k+1)^T}\mathbf{H}^{(l)})$ can be performed in $O(ncq)$.

In general, given that $q$ and $c$ are very small numbers and $m \gg n$, the time complexity is bounded by $O(dmq)$, where $d$ is a constant and $d \ll n$.

**Algorithm 2** LRGNN

**Input:** Graph $\mathcal{G} = (\mathcal{V}, \mathcal{E})$, node features $\mathbf{X}$, adjacency matrix $\mathbf{A}$, Pseudo labels $\bar{\mathbf{Y}}$
**Output:** The final node representation matrix
1: Calculate signed adjacency matrix from pseudo labels using Eq. (10) and Eq. (11).
2: Calculate $\mathbf{H}^{(0)}$ using Eq. (7)
3: Initialize $\mathbf{V}^{(0)}$ and $\mathbf{U}^{(0)}$ using Eq. (22)
4: **for** $k = 0$ to $K - 1$ **do**
5:     Update $\mathbf{U}^{(k+1)}$ using Eq. (25)
6:     Update $\mathbf{V}^{(k+1)}$ using Eq. (27)
7: **end for**
8: Calculate $\mathbf{H}_{out}$ using Eq. (9)
9: **return** $\mathbf{H}_{out}$

### D.3 Convergence Analysis

We establish the following result to justify the design of the surrogate functions.

*Theorem* 4. The objective function Eq.(8) is non-increasing after each update iteration based on algorithm 1 , *i.e.*,

$$F_c(\mathbf{U}^{(k+1)}, \mathbf{V}^{(k+1)}) \leq F_c(\mathbf{U}^{(k+1)}, \mathbf{V}^{(k)}) \leq F_c(\mathbf{U}^{(k)}, \mathbf{V}^{(k)}) \tag{34}$$

*Proof.* This proof is based on the techniques used in [Hastie et al., 2015]. We first prove $F_s(\mathbf{U}^{(k+1)}, \mathbf{V}^{(k)}) \leq F_s(\mathbf{U}^{(k)}, \mathbf{V}^{(k)})$. The design of the surrogate function follows the definition of the Majorization-Minimization (MM) algorithm. An MM algorithm operates by defining a surrogate function that minorizes the objective function. We begin by presenting the following two observations:

$$S_U(\mathbf{U}|\mathbf{U}, \mathbf{V}) = F(\mathbf{U}, \mathbf{V}) \tag{35}$$

and

$$S_U(\mathbf{Z}_U|\mathbf{U}, \mathbf{V}) \geq F(\mathbf{Z}_U, \mathbf{V}) \tag{36}$$

The first equation is easy to derive

$$S_U(\mathbf{U}|\mathbf{U}, \mathbf{V}) = \lambda\|\mathbf{U}\|_F^2 + \lambda\|\mathbf{V}\|_F^2 + \gamma\|\mathbf{H}^{(0)}\mathbf{W}\mathbf{H}^{(0)^T} - \mathbf{U}\mathbf{V}^T\|_F^2 +$$
$$\|\mathbf{M} \odot (\mathbf{U}\mathbf{V}^T - \tilde{\mathbf{A}}) + (\mathbf{1} - \mathbf{M}) \odot (\mathbf{U}\mathbf{V}^T - \mathbf{U}\mathbf{V}^T)\|_F^2$$
$$= \lambda\|\mathbf{U}\|_F^2 + \lambda\|\mathbf{V}\|_F^2 + \gamma\|\mathbf{H}^{(0)}\mathbf{W}\mathbf{H}^{(0)^T} - \mathbf{U}\mathbf{V}^T\|_F^2 + \|\mathbf{M} \odot (\mathbf{U}\mathbf{V}^T - \tilde{\mathbf{A}})\|_F^2 = F(\mathbf{U}, \mathbf{V}) \tag{37}$$

By definition, we have

$$S_U(\mathbf{Z}_U|\mathbf{U}, \mathbf{V}) - F(\mathbf{Z}_U, \mathbf{V}) = \|(\mathbf{1} - \mathbf{M}) \odot (\mathbf{Z}_U\mathbf{V}^T - \mathbf{U}\mathbf{V}^T)\|_F^2 \geq 0. \tag{38}$$

Now we consider the update of $\mathbf{U}$,

$$S_U(\mathbf{U}^{(k+1)}|\mathbf{U}^{(k)}, \mathbf{V}^{(k)}) = \min_{\mathbf{Z}_U} S_U(\mathbf{Z}_U|\mathbf{U}^{(k)}, \mathbf{V}^{(k)}) \leq S_U(\mathbf{U}^{(k)}|\mathbf{U}^{(k)}, \mathbf{V}^{(k)}) \tag{39}$$

Using Eq.(35) and Eq.(36), we have

$$F_s(\mathbf{U}^{(k+1)}, \mathbf{V}^{(k)}) \leq S_U(\mathbf{U}^{(k+1)}|\mathbf{U}^{(k)}, \mathbf{V}^{(k)}) \tag{40}$$

$$S_U(\mathbf{U}^{(k)}|\mathbf{U}^{(k)}, \mathbf{V}^{(k)}) = F_s(\mathbf{U}^{(k)}, \mathbf{V}^{(k)}) \tag{41}$$

Combining these and Eq.(39), we arrive at

$$F_s(\mathbf{U}^{(k+1)}, \mathbf{V}^{(k)}) \leq S_U(\mathbf{U}^{(k+1)}|\mathbf{U}^{(k)}, \mathbf{V}^{(k)}) \leq S_U(\mathbf{U}^{(k)}|\mathbf{U}^{(k)}, \mathbf{V}^{(k)}) = F_s(\mathbf{U}^{(k)}, \mathbf{V}^{(k)}) \tag{42}$$

For simplicity, we rewrite

$$F_s(\mathbf{U}, \mathbf{V}) = \sum_{(i,j)\in\mathcal{E}} s_{i,j}((\mathbf{U}\mathbf{V}^T)_{i,j} - \tilde{\mathbf{A}}_{i,j})^2 + R, \tag{43}$$

where $R$ represents the remaining terms. Since $F_s(\mathbf{U}^{(k+1)}, \mathbf{V}^{(k)}) \leq F_s(\mathbf{U}^{(k)}, \mathbf{V}^{(k)})$, we have

$$\sum_{(i,j)\in\mathcal{E}} s_{i,j}((\mathbf{U}^{(k+1)}\mathbf{V}^{(k)^T})_{i,j} - \tilde{\mathbf{A}}_{i,j})^2 + R' \leq \sum_{(i,j)\in\mathcal{E}} s_{i,j}((\mathbf{U}^{(k)}\mathbf{V}^{(k^T)})_{i,j} - \tilde{\mathbf{A}}_{i,j})^2 + R. \quad (44)$$

Recall that

$$s_{i,j} = \begin{cases} 1, & e_{i,j} = ((\mathbf{U}^{(k)}\mathbf{V}^{(k)^T})_{i,j} - \tilde{\mathbf{A}}_{i,j})^2 < \tau_{i,j} \\ 0, & \text{otherwise} \end{cases} \quad (45)$$

Similarly, we denote $((\mathbf{U}^{(k+1)}\mathbf{V}^{(k)^T})_{i,j} - \tilde{\mathbf{A}}_{i,j})^2$ as $\bar{e}_{i,j}$. We proceed with proving the following inequality

$$min(\bar{e}_{i,j}, \tau) - s_{i,j}\bar{e}_{i,j} \leq min(e_{i,j}, \tau) - s_{i,j}e_{i,j} \quad (46)$$

If $e_{i,j} < \tau_{i,j}$, we have $min(e_{i,j}, \tau) - s_{i,j}e_{i,j} = e_{i,j} - e_{i,j} = 0$, then

$$min(\bar{e}_{i,j}, \tau_{i,j}) - s_{i,j}\bar{e}_{i,j} \leq \bar{e}_{i,j} - \bar{e}_{i,j} = 0 = min(e_{i,j}, \tau_{i,j}) - s_{i,j}e_{i,j}. \quad (47)$$

If $e_{i,j} \geq \tau_{i,j}$, we have $min(e_{i,j}, \tau_{i,j}) - s_{i,j}e_{i,j} = \tau_{i,j}$, then

$$min(\bar{e}_{i,j}, \tau_{i,j}) - s_{i,j}\bar{e}_{i,j} = min(\bar{e}_{i,j}, \tau_{i,j}) \leq \tau_{i,j} = min(e_{i,j}, \tau_{i,j}) - s_{i,j}e_{i,j}. \quad (48)$$

Therefore, we can say that Eq.(46) holds. By summing over Eq.(44) and Eq.(46) in two sides, we reach the following inequality

$$\sum_{(i,j)\in\mathcal{E}} min(((\mathbf{U}^{(k+1)}\mathbf{V}^{(k)^T})_{i,j} - \tilde{\mathbf{A}}_{i,j})^2, \tau_{i,j}) + R' \leq \sum_{(i,j)\in\mathcal{E}} min(((\mathbf{U}^{(k)}\mathbf{V}^{(k)^T})_{i,j} - \tilde{\mathbf{A}}_{i,j})^2, \tau_{i,j}) + R, \quad (49)$$

which is equivalent to

$$F_c(\mathbf{U}^{(k+1)}, \mathbf{V}^{(k)}) \leq F_c(\mathbf{U}^{(k)}, \mathbf{V}^{(k)}). \quad (50)$$

The proof of $F_c(\mathbf{U}^{(k+1)}, \mathbf{V}^{(k+1)}) \leq F_c(\mathbf{U}^{(k+1)}, \mathbf{V}^{(k)})$ can be accomplished the same way, and we will omit it.

$\square$

# E  Related Work

In this section, we discuss relevant work that addresses the heterophily challenge. Abu-El-Haija et al. [2019] acknowledges the limitations of current GNNs in learning on graphs with heterophily and proposes to exploit higher-order information by aggregating multi-hop neighborhoods. The authors of Zhu et al. [2020] further identified several effective designs and provided theoretical justifications. Chien et al. [2021] generalizes the PageRank and proposes GPR-GNN that performs well under heterophily. FAGCN [Bo et al., 2021] utilizes a self-gating attention mechanism to adaptively learn the proportion of low-frequency and high-frequency signals. Later, WRGAT [Suresh et al., 2021] transforms the original graph into a new multi-relational one with a higher homophily ratio. The authors of Yan et al. [2021] regard oversmoothing and heterophily as two sides of the same coin. They suggest addressing these two issues via degree correction and signed message. LINKX [Lim et al., 2021] first embeds node features and graph topology separately and then combines them with MLPs. Recently, GloGNN [Li et al., 2022] proposes to leverage global homophily and derives a coefficient matrix that optimizes a well-designed objective function. Zheng et al. [2022] provides a comprehensive survey on GNNs for heterophilous graphs.

**GloGNN**. A closely related work is the GloGNN [Li et al., 2022], which defines a coefficient matrix as

$$\mathbf{Z}_*^{(l)} = \underset{\mathbf{Z}^{(l)}\in\mathbb{R}^{n\times n}}{argmin} \|\mathbf{H}^{(l)} - (1-\beta)\mathbf{Z}^{(l)}\mathbf{H}^{(l)} - \beta\mathbf{H}^{(0)}\|_F^2 + \gamma\|\mathbf{Z}^{(l)} - \mathbf{A}_{\text{GCN}}^k\|_F^2 + \lambda\|\mathbf{Z}^{(l)}\|_F^2 \quad (51)$$

Here we point out several major differences between our LRGNN and GloGNN. The coefficient matrix in LRGNN is a low-rank matrix that aligns with the weak balance theory, reflecting the low-rank structures of complete graphs. LRGNN recovers the label relationship matrix using matrix approximation, whereas GloGNN regularizes the missing edges to be zero by including a $\|\mathbf{Z}^{(l)} - \mathbf{A}_{\text{GCN}}^k\|_F^2$ term. In GloGNN, the adjacency matrix is the symmetric normalized adjacency matrix (or its powers) utilized in vanilla GCN. It has uniform and positive edge weights. In contrast,

for LRGNN, the similarity matrix is generated using pseudo labels with the allowance for negative weights. Furthermore, LRGNN uses an element-wise capped norm to identify the outliers and restrict their contribution to the total loss. Lastly, the first term of GloGNN involves subspace clustering, whereas LRGNN comprises the attention term.

To conclude, LRGNN is a low-rank matrix approximation model that utilizes an attention term to further improve performance. On the other hand, GloGNN is a subspace clustering model that incorporates an adjacency matrix regularized term to enhance its performance.

# F   Datasets

Here we briefly introduce the datasets used in our experiments. In particular, these datasets span various domains and edge homophily.

**Cora, Citeseer and Pubmed** are citation networks where nodes represent scientific papers and edges are citation relationships. Node features are bag-of-words representations and each label represents the field that the paper belongs to.

**Actor** is a co-occurrence network generated from the film-director-actor-writer network, where node features are bag-of-words representations of the Wikipedia pages of actors. Edges symbolize the two actors' co-occurrence on the same web page.

**Cornell, Texas and Wisconsin** are collected as part of CMU WebKB project. In these datasets, nodes are university web pages and edges are hyperlinks between these pages.

**Chameleon and Squirrel** are two networks of web pages on Wikipedia regarding animals. Node features are bag-of-words representations of nouns in the respective pages. The task is to classify pages into five categories based on the average traffic they received.

**Synthetic graphs** are controlled by the node-level homophily ratio and the average degree. Specifically, a random graph contains $n$ nodes per class and c classes, with two probabilities $p_{in}$ and $p_{out}$, where $p_{in}$ corresponds to the probability of forming a intra-class edge, and $p_{out}$ corresponds to the probability of forming a inter-class edge. We choose $p_{in}$ and $p_{out}$ by $p_{in} + (c-1) \cdot p_{out} = \delta$, and the average degree of the random graph is $d_{avg} = n\delta$. We set $n$ to 500, $c$ to 5, and select $d_{avg}$ from $\{0.5, 5, 20\}$, $p_{in}$ from $\{0.1\delta, 0.3\delta, 0.5\delta, 0.7\delta, 0.9\delta\}$. The node features are sampled from Gaussian distributions where the centers of clusters are vertices of a hypercube. Nodes are randomly split into $(10\%/45\%/45\%)$ for training/validation/testing. Note that $p_{in} = 0.9\delta$ indicates strong homophily and $p_{in} = 0.1\delta$ corresponds to strong heterophily.

**arXiv-year** is a directed subgraph of ogbn-arXiv, where nodes are arXiv papers and edges represent the citation relations. Node features is constructed by taking the averaged word2vec embedding vectors of tokens contained in both the title and abstract of papers.

**Penn94** is a subgraph extracted from Facebook whose nodes are students. Node features include major, second major/minor, dorm/house, year and high school. Students' genders is used for nodes' labels.

**genius** is a subnetwork extracted from genius.com, which is a website for crowdsourced annotations of song lyrics. In the graph, nodes represent users and edges connect users that follow each other. Node features include expertise scores, counts of contributions, roles held by users, etc. Users who are more likely to be spam users are marked with a "gone" label on the site. The task is to predict whether a user is marked with "gone".

# G   SignedGNNs

**SignedGCNs.**   Since the over-smoothing problem is partly caused by the coupling of neighborhood aggregation and feature transformation as pointed out in Liu et al. [2020], we remove the feature transformation of the middle layer. Further, to avoid dimension explosion, the vector concatenation $\|$ is actually implemented with the average sum. In each layer, SignedGCNs maintain a "friend" and an "enemy" representation for each node. At the first layer, it constructs the "friend" and the "enemy"

representation as,

$$h_{i,f}^{(1)} = [\frac{1}{|\mathcal{N}_i^+|} \sum_{j \in \mathcal{N}_i^+} h_j^{(0)} || h_i^{(0)}], h_{i,e}^{(1)} = [\frac{1}{|\mathcal{N}_i^-|} \sum_{k \in \mathcal{N}_i^-} h_k^{(0)} || h_i^{(0)}], \tag{52}$$

where $\mathcal{N}_i^+$ and $\mathcal{N}_i^-$ are the homophilous and heterophilous neighbors, respectively. The subscript "f" and "e" denote "friend" and "enemy" representations, respectively. Here the "friend" representation is actually the average of homophilous raw features and ego features, so it only contains homophilous information. Given $l \geq 2$, it updates the "friend" representation using "f-f-f" and "e-e-f",

$$h_{i,f}^{(l)} = [\frac{1}{|\mathcal{N}_i^+|} \sum_{j \in \mathcal{N}_i^+} h_{j,f}^{(l-1)} || \frac{1}{|\mathcal{N}_i^-|} \sum_{k \in \mathcal{N}_i^-} h_{k,e}^{(l-1)} || h_{i,f}^{(l-1)}], \tag{53}$$

and the "enemy" representation using "e-f-e" and "f-e-e",

$$h_{i,e}^{(l)} = [\frac{1}{|\mathcal{N}_i^+|} \sum_{j \in \mathcal{N}_i^+} h_{j,e}^{(l-1)} || \frac{1}{|\mathcal{N}_i^-|} \sum_{k \in \mathcal{N}_i^-} h_{k,f}^{(l-1)} || h_{i,e}^{(l-1)}], \tag{54}$$

Here the problem arises. $h_{k,e}^{(l-1)}, k \in \mathcal{N}_i^-$, could be a heterophilous node representation under a multi-class classification scenario. So at the later layer, heterophilous information might also be encoded in the "friend" representation. As a result, heterophilous and homophilous information is mixed in the "friend" representation.

**WB-SignedGCN.** We retain the "enemy" representation and modify the "friend" representation as

$$h_{i,f}^{(l)} = [\frac{1}{|\mathcal{N}_i^+|} \sum_{j \in \mathcal{N}_i^+} h_{j,f}^{(l-1)} || h_{i,f}^{(l-1)}], \tag{55}$$

where the assumption "e-e-f" has been eliminated, which gives a weak balance version of SignedGCN termed WB-SignedGCN.

# H  Proof of Theorem 2

*Theorem* 2. Consider that we apply a multiple-layer signed GNN on a triad. Assume that each coefficient is independent of other coefficients and the probability that the model can precisely predict the sign of each coefficient is $p$, namely $Pr(\text{sign}(\alpha_{i,j}^{(l)}) = y_{i,j}) = p$. Also, assume that all the self-coefficients are positive. The probability that at least one of $\bar{\alpha}_{i,j}^{(L)}$ and $\bar{\alpha}_{j,k}^{(L)}$ are correct in sign and $< i, j, k >$ being balanced is given by $p_b$. Then, $p_b$ is monotonically increasing concerning $p$. Especially, if $p = 1$, the triad is always balanced.

*Proof.* A multiple-layer model can be formulated as

$$\mathbf{H}^{(L)} = \mathcal{A}^{(L)} \cdots \mathcal{A}^{(1)} \mathbf{H}^{(0)}, \tag{56}$$

where $\mathcal{A}^{(l)}$ are matrices with $\mathcal{A}_{i,j}^{(l)} = \alpha_{i,j}^{(l)}$ denoting the coefficient $v_i$ gives to $v_j$ in the $l$-th layer. Let $\bar{\mathcal{A}}^{(l)} = \mathcal{A}^{(l)} \cdots \mathcal{A}^{(1)}$, then the output of a $l$-layer model can be rewritten as $\mathbf{H}^{(l)} = \bar{\mathcal{A}}^{(l)} \mathbf{H}^{(0)}$. Also, denote $\bar{\mathcal{A}}_{i,j}^{(l)}$ by $\bar{\alpha}_{i,j}^{(l)}$, such that $\mathbf{H}_{i,:}^{(l)} = \sum_j \bar{\alpha}_{i,j}^{(l)} \mathbf{H}_{j,:}^{(0)}$. Hence, $\bar{\alpha}_{i,j}^{(l)}$ can be regarded as the final learned signed weight $v_i$ gives to $v_j$ for a $l$-layer model. Consider $L = 2$

$$\bar{\alpha}_{i,j}^{(2)} = \alpha_{i,i}^{(2)} \alpha_{i,j}^{(1)} + \alpha_{i,j}^{(2)} \alpha_{j,j}^{(1)}. \tag{57}$$

$$\bar{\alpha}_{j,k}^{(2)} = \alpha_{j,j}^{(2)} \alpha_{j,k}^{(1)} + \alpha_{j,k}^{(2)} \alpha_{k,k}^{(1)}. \tag{58}$$

$$\bar{\alpha}_{i,k}^{(2)} = \alpha_{i,j}^{(2)} \alpha_{j,k}^{(1)}. \tag{59}$$

case 1: $y_{i,j} = 1$ and $y_{j,k} = 1$.

$$p_b = Pr(\bar{\alpha}_{i,j}^{(2)} > 0, \bar{\alpha}_{j,k}^{(2)} > 0, \bar{\alpha}_{i,k}^{(2)} > 0) + Pr(\bar{\alpha}_{i,j}^{(2)} > 0, \bar{\alpha}_{j,k}^{(2)} < 0, \bar{\alpha}_{i,k}^{(2)} < 0)$$
$$+ Pr(\bar{\alpha}_{i,j}^{(2)} < 0, \bar{\alpha}_{j,k}^{(2)} > 0, \bar{\alpha}_{i,k}^{(2)} < 0) \tag{60}$$

By independence,

$$Pr(\bar{\alpha}_{i,j}^{(2)} > 0, \bar{\alpha}_{j,k}^{(2)} > 0, \bar{\alpha}_{i,k}^{(2)} > 0) = Pr(\bar{\alpha}_{i,j}^{(2)} > 0) \cdot Pr(\bar{\alpha}_{j,k}^{(2)} > 0) \cdot Pr(\bar{\alpha}_{i,k}^{(2)} > 0) \quad (61)$$

We have

$$Pr(\bar{\alpha}_{i,j}^{(2)} > 0) = Pr(\alpha_{i,j}^{(1)} > 0) \cdot Pr(\alpha_{i,j}^{(2)} > 0) + Pr(\alpha_{i,j}^{(1)} > 0) \cdot Pr(\alpha_{i,j}^{(2)} < 0) \cdot$$
$$Pr(|\alpha_{i,i}^{(2)}\alpha_{i,j}^{(1)}| > |\alpha_{i,j}^{(2)}\alpha_{j,j}^{(1)}|) + Pr(\alpha_{i,j}^{(1)} < 0) \cdot Pr(\alpha_{i,j}^{(2)} > 0) \cdot Pr(|\alpha_{i,i}^{(2)}\alpha_{i,j}^{(1)}| < |\alpha_{i,j}^{(2)}\alpha_{j,j}^{(1)}|)$$
$$= p^2 + p(1-p) \cdot Pr(|\alpha_{i,i}^{(2)}\alpha_{i,j}^{(1)}| > |\alpha_{i,j}^{(2)}\alpha_{j,j}^{(1)}|) + (1-p)p \cdot Pr(|\alpha_{i,i}^{(2)}\alpha_{i,j}^{(1)}| < |\alpha_{i,j}^{(2)}\alpha_{j,j}^{(1)}|)$$
$$= p^2 + p(1-p) \cdot (Pr(|\alpha_{i,i}^{(2)}\alpha_{i,j}^{(1)}| < |\alpha_{i,j}^{(2)}\alpha_{j,j}^{(1)}|) + Pr(|\alpha_{i,i}^{(2)}\alpha_{i,j}^{(1)}| > |\alpha_{i,j}^{(2)}\alpha_{j,j}^{(1)}|)) = p^2 + p(1-p) = p$$
$$(62)$$

In the same way, we can obtain $Pr(\bar{\alpha}_{j,k}^{(2)} > 0)$.

$$Pr(\bar{\alpha}_{j,k}^{(2)} > 0) = Pr(\alpha_{j,k}^{(1)} > 0) \cdot Pr(\alpha_{j,k}^{(2)} > 0) + Pr(\alpha_{j,k}^{(1)} > 0) \cdot Pr(\alpha_{j,k}^{(2)} < 0) \cdot$$
$$Pr(|\alpha_{j,j}^{(2)}\alpha_{j,k}^{(1)}| > |\alpha_{j,k}^{(2)}\alpha_{k,k}^{(1)}|) + Pr(\alpha_{j,k}^{(1)} < 0) \cdot Pr(\alpha_{j,k}^{(2)} > 0) \cdot Pr(|\alpha_{j,j}^{(2)}\alpha_{j,k}^{(1)}| < |\alpha_{j,k}^{(2)}\alpha_{k,k}^{(1)}|)$$
$$= p^2 + p(1-p) \cdot Pr(|\alpha_{j,j}^{(2)}\alpha_{j,k}^{(1)}| > |\alpha_{j,k}^{(2)}\alpha_{k,k}^{(1)}|) + (1-p)p \cdot Pr(|\alpha_{j,j}^{(2)}\alpha_{j,k}^{(1)}| < |\alpha_{j,k}^{(2)}\alpha_{k,k}^{(1)}|)$$
$$= p + p(1-p) = p \quad (63)$$

Next, we derive $Pr(\bar{\alpha}_{i,k}^{(2)} > 0)$

$$Pr(\bar{\alpha}_{i,k}^{(2)} > 0) = Pr(\alpha_{i,j}^{(2)} > 0) \cdot Pr(\alpha_{j,k}^{(1)} > 0) + Pr(\alpha_{i,j}^{(2)} < 0) \cdot Pr(\alpha_{j,k}^{(1)} < 0) = p^2 + (1-p)^2$$
$$(64)$$

Therefore,

$$Pr(\bar{\alpha}_{i,j}^{(2)} > 0) \cdot Pr(\bar{\alpha}_{j,k}^{(2)} > 0) \cdot Pr(\bar{\alpha}_{i,k}^{(2)} > 0) = p^2(p^2 + (1-p)^2) \quad (65)$$

$$Pr(\bar{\alpha}_{i,j}^{(2)} > 0, \bar{\alpha}_{j,k}^{(2)} < 0, \bar{\alpha}_{i,k}^{(2)} < 0) = Pr(\bar{\alpha}_{i,j}^{(2)} > 0) \cdot (1 - Pr(\bar{\alpha}_{j,k}^{(2)} > 0)) \cdot (1 - Pr(\bar{\alpha}_{i,k}^{(2)} > 0)) = 2p^2(1-p)^2$$
$$(66)$$

$$Pr(\bar{\alpha}_{i,j}^{(2)} < 0, \bar{\alpha}_{j,k}^{(2)} > 0, \bar{\alpha}_{i,k}^{(2)} < 0) = (1 - Pr(\bar{\alpha}_{i,j}^{(2)} > 0)) \cdot Pr(\bar{\alpha}_{j,k}^{(2)} > 0) \cdot (1 - Pr(\bar{\alpha}_{i,k}^{(2)} > 0)) = 2p^2(1-p)^2$$
$$(67)$$

$$p_b = 2p^2(1-p)^2 + 2p^2(1-p)^2 + p^2(p^2 + (1-p)^2) = p^4 + 5p^2(1-p)^2 \quad (68)$$

Let $f(p) = p^4 + 5p^2(1-p)^2$

$$f(p)' = 24p^3 - 30p^2 + 10p, \quad \frac{f(p)'}{p} = 24p^2 - 30p + 10 > 0 \quad (30^2 - 4*24*10 < 0) \quad (69)$$

Since $p > 0$, we arrive at $f(p)' > 0$, then $p_b$ is monotonically increasing. It is easy to verify that $f(0) = 0$ and $f(1) = 1$.

case 2: $y_{i,j} = 1$ and $y_{j,k} = -1$.

$Pr(\bar{\alpha}_{i,j}^{(2)} > 0) = p$ and $Pr(\bar{\alpha}_{j,k}^{(2)} < 0) = p$ can be easily derived in the same way

$$p_b = Pr(\bar{\alpha}_{i,j}^{(2)} > 0, \bar{\alpha}_{j,k}^{(2)} < 0, \bar{\alpha}_{i,k}^{(2)} < 0) + Pr(\bar{\alpha}_{i,j}^{(2)} < 0, \bar{\alpha}_{j,k}^{(2)} < 0, \bar{\alpha}_{i,k}^{(2)} > 0) + Pr(\bar{\alpha}_{i,j}^{(2)} > 0, \bar{\alpha}_{j,k}^{(2)} > 0, \bar{\alpha}_{i,k}^{(2)} > 0).$$

We first derive

$$Pr(\bar{\alpha}_{i,k}^{(2)} < 0) = Pr(\bar{\alpha}_{i,j}^{(2)} > 0)Pr(\bar{\alpha}_{j,k}^{(1)} < 0) + Pr(\bar{\alpha}_{i,j}^{(2)} < 0)Pr(\bar{\alpha}_{j,k}^{(2)} > 0) = p^2 + (1-p)^2. \quad (70)$$

Then, we have

$$Pr(\bar{\alpha}_{i,j}^{(2)} > 0, \bar{\alpha}_{j,k}^{(2)} < 0, \bar{\alpha}_{i,k}^{(2)} < 0) = Pr(\bar{\alpha}_{i,j}^{(2)} > 0) \cdot Pr(\bar{\alpha}_{j,k}^{(2)} < 0) \cdot Pr(\bar{\alpha}_{i,k}^{(2)} < 0) = p^2(p^2 + (1-p)^2).$$
$$(71)$$

Since

$$Pr(\bar{\alpha}_{i,j}^{(2)} < 0, \bar{\alpha}_{j,k}^{(2)} < 0, \bar{\alpha}_{i,k}^{(2)} > 0) = Pr(\bar{\alpha}_{i,j}^{(2)} < 0) \cdot Pr(\bar{\alpha}_{j,k}^{(2)} < 0) \cdot Pr(\bar{\alpha}_{i,k}^{(2)} > 0) = 2p^2(1-p)^2.$$
$$(72)$$

| Hyper-parameter | Range |
|---|---|
| learning rate | $\{0.01, 0.005, 0.02\}$ |
| weight decay | $\{5e-3, 5e-4, 5e-5\}$ |
| dropout | $[0, 0.7]$ |
| early stopping | $\{40, 100, 200\}$ |
| $\beta$ | $[0.6, 0.9]$ |
| $\mu$ | $[0.1, 0.9]$ |
| $\delta$ | $[0, 0.9]$ |
| $\gamma$ | $\{0.0001, 0.001, 0.002, 0\}$ |
| $\lambda$ | $[0.01, 0.02, 0.05, 0.001]$ |
| K | $\{1, 2\}$ |
| Estimator | $\{\text{GCN, MLP}\}$ |

Table 6: Search space for hyper-parameters.

and

$$Pr(\bar{\alpha}_{i,j}^{(2)} > 0, \bar{\alpha}_{j,k}^{(2)} > 0, \bar{\alpha}_{i,k}^{(2)} > 0) = Pr(\bar{\alpha}_{i,j}^{(2)} > 0) \cdot Pr(\bar{\alpha}_{j,k}^{(2)} > 0) \cdot Pr(\bar{\alpha}_{i,k}^{(2)} > 0) = 2p^2(1-p)^2. \tag{73}$$

We obtain

$$p_b = 2p^2(1-p)^2 + 2p^2(1-p)^2 + p^2(p^2 + (1-p)^2) = p^4 + 5p^2(1-p)^2 \tag{74}$$

Therefore, case 2 gives the same probability as case 1. Given that the case $y_{i,j} = -1, y_{j,k} = -1$ is symmetric to case 1, and $y_{i,j} = -1, y_{j,k} = 1$ is symmetric to case 2, we can prove that $p_b$ is always monotonically increasing.

We next consider the situation of $L = 3$. Note that $\bar{\mathcal{A}}^{(3)} = \mathcal{A}^{(3)}\bar{\mathcal{A}}^{(2)}$ Hence, we have

$$\bar{\alpha}_{i,j}^{(3)} = \alpha_{i,i}^{(3)}\bar{\alpha}_{i,j}^{(2)} + \alpha_{i,j}^{(3)}\bar{\alpha}_{j,j}^{(2)}. \tag{75}$$

$$\bar{\alpha}_{j,k}^{(3)} = \alpha_{j,j}^{(3)}\bar{\alpha}_{j,k}^{(2)} + \alpha_{j,k}^{(3)}\bar{\alpha}_{k,k}^{(2)}. \tag{76}$$

$$\bar{\alpha}_{i,k}^{(3)} = \alpha_{i,j}^{(3)}\bar{\alpha}_{j,k}^{(2)}. \tag{77}$$

We have proved that $Pr(\bar{\alpha}_{i,j}^{(2)} > 0) = Pr(\alpha_{i,j}^{(1)} > 0)$ and $Pr(\bar{\alpha}_{j,k}^{(2)} > 0) = Pr(\alpha_{j,k}^{(1)} > 0)$. By assumption, $\bar{\alpha}_{j,j}^{(2)} > 0$ and $\bar{\alpha}_{k,k}^{(2)} > 0$. Therefore, the case $L = 3$ degenerates to the case $L = 2$. We can prove in the same way that the probability $p_b$ is monotonically increasing with respect to $p$. By further proving for the case $L = 4, 5 \cdots$, we complete the proof.

$\square$

# I  Experimental Setup

We implement LRGNN with Pytorch. We ran our experiments on an Nvidia A100 GPU with 40GB of memory. For real-world graphs, we use 10 random splits ($48\%, 32\%20\%$ for training/validation/testing) provided by Pei et al. [2020] and available from Pytorch Geometry [Fey and Lenssen, 2019]. For real-world datasets, since the results of these baseline methods on these benchmark datasets are public, we directly report these results. For the results on synthetic graphs, we run the baseline methods using the codes released by their authors and fine-tune the hyper-parameters based on the validation set. We perform a grid search to tune hyper-parameters based on the validation set, as shown in Table 6.