# OpenReview forum: "Predicting Global Label Relationship Matrix for Graph Neural Networks under Heterophily"
_NeurIPS.cc/2023/Conference — NeurIPS 2023 poster_

### Official Review · Reviewer_B8tM · 2023-07-03

**Soundness:** 3 good
**Presentation:** 3 good
**Contribution:** 3 good
**Rating:** 5
**Confidence:** 4

**Summary:**

The paper proposes a Low-Rank Graph Neural Network (LRGNN) that can model both homophilous and heterophilous graphs via signed propogation (two nodes with the same class label share a positive edge and with different class share a negative edge). The authors predict the label relationship matrix by solving a robust low-rank matrix approximation problem, which results in a block diagonal structure and varying distributions of within-class and between-class entries.


**Strengths:**

Clarity: The paper is generally easy to follow.

Novelty: The authors propose a low-rank matrix approximation technique for predicting the label relationship matrix and enhancing the representation power of nodes. This technique is designed to utilize the low-rank properties of weakly-balanced graphs.

Experiments: The authors have conducted extensive experiments on both synthetic and real-world graphs to evaluate the proposed method. The results indicate that LRGNN outperforms other baseline methods on both synthetic and real-world graphs.

Presentation: The paper is well-structured and many visualization analysis are provided to better assess the qualitative and quantitive analysis of the proposed model.



**Weaknesses:**

Assumption of Uniform Distribution: The main theoretical support for the approach assumes a uniform distribution of node degrees. However, in practice, node degree distributions frequently follow a power-law distribution that is non-uniform. Although the main results of this paper are obtained using real-world graphs with non-uniform node degree distributions, it is unclear how this affects the theoretical results.

Dependence on Pseudo Labels: It is mentioned that LRGNN’s performance is affected by the accuracy of the generated pseudo labels. Although the supplementary material provides an experiment that indicates LRGNN demonstrates a non-sensitive response to this impact, further exploration and discussion are needed to fully investigate the theoretical implications. Additionally, the motivation of using signed labels for edges is not strong enough becuase 1) only applies for homogenous graphs. 2) When the # of class labels is high, such strategy may lose a lot class-related information so it may not generalize.

**Questions:**

N/A

---

> ### Author Rebuttal · Authors · 2023-08-08
>
> We appreciate the reviewer's positive acknowledgment  on the clarity, novel, experiments, and presentation of our paper. For the weakness part, we have indeed discussed these limitations in the last section of our paper and devote much efforts to support the robustness of LRGNN by presenting extensive designated experiments in our experiment section as well as the Appendix.
>
> Uniform Distribution:  The real-world datasets considered in our research exhibit non-uniform distributions, yet our experimental results validate the strong performance of our model on such datasets. In Appendix A, we have provided a detailed discussion and conducted an experiment to empirically demonstrate the effectiveness of low-rank matrix factorization (LRMF) in accurately predicting the label relationship matrix, despite the non-uniform distribution of observed entries.
>
> Dependence on Pseudo Labels: We believe LRGNN does not rely on accurate pseudo labels. In fact, the weakest estimator used to generate pseudo labels is the one for the Squirrel, which has an accuracy of about 50%, while LRGNN achieves an impressive 74.38% accuracy on Squirrel, surpassing the runner-up score significantly. Thus, LRGNN demonstrates to be effective even when the generated pseudo labels are less reliable. Moreover, the experimental results in the Appendix indicate that the performance of LRGNN endures only a minor impact when the pseudo labels are corrupted by random noises.
>
> We recognize the significance of the theoretical implications that arise from the shape of the distribution and the accuracy of pseudo labels.  We plan to dedicate our future work to explore the theoretical aspects.
>
> In summary, we appreciate the reviewer for acknowledging the strengths of our paper include the clarity, novelty, experiments, and presentation. For the limitation mentioned in our paper, we will make further improvement upon our recent trials in the future work.

---

> ### Comment · Area_Chair_xttR · 2023-08-17
> **About Authors' Reply**
>
> Dear reviewer B8tM,
>
> Would you mind to check authors' reply and indicate whether you'd like to change the score or need more discussion?
>
> AC

---

### Official Review · Reviewer_jP2E · 2023-07-06

**Soundness:** 3 good
**Presentation:** 3 good
**Contribution:** 2 fair
**Rating:** 5
**Confidence:** 4

**Summary:**

This paper introduces sign links to model both the homophilous and heterophilous relationships in the global graph, thus extending the applicability of GNNs beyond heterophily. Additionally, the paper leverages the weak balance theory to support the existence of a global low-rank structure in the signed graph and formulates sign link prediction as a robust low-rank matrix approximation problem. Experimental results demonstrate the effectiveness and performance improvement of the proposed approach.

**Strengths:**

Clarity:
1. The submission demonstrates clear writing, readability, and a well-organized structure.
2. The supplementary material provides additional details that enhance the reproducibility of the results.

Quality:
1. The claims put forth in the submission are strongly backed by theoretical and empirical analyses.
2. The inclusion of the structural balance theory and conducting accuracy comparison, ablation, and visualization experiments significantly bolsters the overall quality of the submission.

**Weaknesses:**

Originality
1. The originality of the contributions can be strengthened by explaining the differences and comparing them with several relevant existing works, such as FAGCN [1], HOG-GCN [2], etc.
2. In particular, the description in the paper about how the proposed signed GNN can better work for heterophily problems is insufficient.

Significance
1. Most heterophily datasets considered in the paper have issues (e.g., small scale, train-test data leakage, etc.) as pointed out by recent publications on high-quality heterophily datasets [3].
2. Strengthening the significance of the contributions can be achieved by incorporating important references and conducting experimental comparisons.

[1]Bo, D., Wang, X., Shi, C., & Shen, H. (2021). Beyond low-frequency information in graph convolutional networks. In AAAI, 35, 3950-3957.

[2]Wang, T., Jin, D., Wang, R., He, D., & Huang, Y. (2022). Powerful graph convolutional networks with adaptive propagation mechanism for homophily and heterophily. In AAAI, 36, 4210-4218.

[3]Platonov, O., Kuznedelev, D., Diskin, M., Babenko, A., & Prokhorenkova, L. (2023). A critical look at the evaluation of GNNs under heterophily: Are we really making progress?. In ICLR.

**Questions:**

1. The idea behind LRGNN bears similarities to FAGCN; however, the author fails to provide a detailed explanation of the differences between them.
2. There is a lack of experimental comparisons with state-of-the-art (SOTA) models such as FAGCN [1], DMP [4], and Ordered GNN [5].
3. The paper lacks experimental comparisons with other popular large-scale datasets, as mentioned in [6], such as pokec, snap-patents, and twitch-gamers.
4. Figure 3 solely displays the recovery loss on heterophily graphs. It would be valuable to include the results on homophily graphs as well.

[4]Yang, L., Li, M., Liu, L., Wang, C., Cao, X., & Guo, Y. (2021). Diverse message passing for attribute with heterophily. In NeurIPS, 34, 4751-4763.

[5]Song, Y., Zhou, C., Wang, X., & Lin, Z. (2023). Ordered GNN: Ordering message passing to deal with heterophily and over-smoothing. In ICLR.

[6]Lim, D., Hohne, F., Li, X., Huang, S. L., Gupta, V., Bhalerao, O., & Lim, S. N. (2021). Large scale learning on non-homophilous graphs: New benchmarks and strong simple methods. In NeurIPS, 34, 20887-20902.

---------
Update: Most of the above concerns have been addressed.

---

> ### Author Rebuttal · Authors · 2023-08-08
>
> Thank you for your valuable suggestions for improving our paper and for appraising the quality and clarity of our paper. Here are our responses to your questions, and we put the tables in the uploaded pdf file.
>
> **Q1 On the differences between LRGNN and existing models.**
>
> **R1** Thanks for the suggestion, and we would like to discuss the limitation concerning the expressive power of FAGCN. FAGCN derives the propagation weights using neural networks, while LRGNN is based on traditional matrix completion.  In FAGCN, the propagation weight for edge $(i,j)$ is computed as
>
> $\alpha_{i,j} = tanh(g^T [h_i || h_j]) = tanh (g_1^T h_i + g_2^Th_j), g = [g_1||g_2]$
>
> This attention function has been proven to be static, as shown in recent research [1].
> Since tanh is a monotonic function, for $\forall i,j,m,n$,  if $\alpha_{i,m}>\alpha_{i,n}$, we have $g_1^T h_i + g_2^Th_m > g_1^T h_i + g_2^Th_n$. Therefore, $g_2^Th_m>g_2^Th_n$, which indicates $g_1^T h_j+ g_2^Th_m > g_1^T h_j + g_2^Th_n$ and $\alpha_{j,m}>\alpha_{j,n}$. In short, for FAGCN, if we observe $\alpha_{i,m}>\alpha_{i,n}$, then we know $\alpha_{j,m}>\alpha_{j,n}$, irrespective of the label relationships between $j$ and $m$, or $j$ and $n$.  Hence, the ranking of the attention scores is unconditioned on the query node, $i$ and $j$ in this case.
>
> In contrast, our visualization results (Fig. 4) demonstrate that For LRGNN, the within-class, and between-class weights are positioned on opposite sides of the zero.  As discussed in Sec. 3, FAGCN is a standard signed GNN based on structural balance theory, while LRGNN is a naturally weakly-balanced model as it aggregates all nodes’ representations for a node in one-hop propagation. This together explains LRGNN's superiority over FAGCN.
>
> The signed feature propagation has not been explicitly modeled **in HOG-GCN**. This is crucial because negative propagation weights are necessary to push away node pairs with different labels in the embedding space. Additionally, HOG-GCN amplifies the neighborhood of nodes by explicitly computing $A^k$. However, this calculation greatly increases the time complexity. For instance, when $k=3$, the time complexity of HOG-GCN becomes cubic, while the time complexity of LRGNN is linear to the # of edges.
>
> [1] Brody, Shaked, Uri Alon, and Eran Yahav. "How attentive are graph attention networks?." ICLR 2022.
>
> **Q2 There is a lack of experimental comparisons with state-of-the-art (SOTA) models...**
>
> **R2** We **included Ordered GNN** as a baseline model in our experiment section for performance comparison. Please refer to line 272, table 2, and table 3 (**abbreviated as OGNN** in the table otherwise the page will be overwhelmed by the table). Unfortunately, it seems that the authors of DMP and HOG-GCN did not make their source codes publicly available, as we could not find any relevant implementations on GitHub or any links to the source codes in their papers. Additionally, the dataset split used in DMP differs from our own, and the experimental results provided in the HOG-GCN paper only cover a subset of the datasets used in our paper.  Given these limitations, we can only implement FAGCN and directly use the available results from the HOG-GCN paper for comparison. Please see the pdf file for the results.
>
> **Q3 The limitations of the datasets considered in the paper**
>
> **R3** Actually, we read the paper [2] during its reviewing phase and highly valued its insights and contributions to the community. Nevertheless, we selected these datasets as they are currently the most widely utilized by the community, despite their known limitations. For example, a recent paper [3] accepted at ICML and the most recent papers [4][5] published on arXiv adopted these datasets for their experiments. We duly appreciate the reviewer's efforts in advancing the evaluation of GNNs under heterophily. We provide the results on the datasets from [2] for your reference. Note that the results of baselines are directly taken from [2].
>
> As indicated by [2], classic GNNs generally perform better than heterophily-specific models on these datasets. The performance of LRGNN is comparable to FSGNN and surpasses other heterophily-specific models.
>
> [2] Platonov, Oleg, et al. "A critical look at the evaluation of GNNs under heterophily: are we really making progress?." ICLR 2023.
>
> [3] Zheng, Yizhen, et al. "Finding the Missing-half: Graph Complementary Learning for Homophily-prone and Heterophily-prone Graphs." ICML 2023.
>
> [4] Wang, Junfu, et al. "Heterophily-Aware Graph Attention Network." arXiv preprint arXiv:2302.03228 (2023).
>
> [5] Yang, Wenhan, and Baharan Mirzasoleiman. "Contrastive Learning under Heterophily." arXiv preprint arXiv:2303.06344 (2023).
>
> **Q4 The paper lacks experimental comparisons with other popular large-scale datasets...**
>
> **R4** We have included three out of six large-scale datasets released by [6] for evaluation. However, the remaining datasets, pokec, snap-patents, and twitch-gamers are too large, and such a huge scale presents significant computational challenges for our machines. For instance, pokec consists of over 1.6 million nodes and 30 million edges. We acknowledge the significance of evaluating a new method using large-scale datasets and understand the concern expressed by the reviewer. We are sorry that, unfortunately, due to limited resources, we are unable to conduct experiments on pokec and snap-patents. Thus, we only present the results on the twitch-gamers dataset, which is processed using CPUs. We will consider other datasets in our future work once more computational resources are available.
>
> [6] Lim, Derek, et al. "Large scale learning on non-homophilous graphs: New benchmarks and strong simple methods." NeurIPS 2022.
>
> **Q5 Figure 3 solely displays the recovery loss on heterophily graphs...**
>
> **R5** The visualization results on the homophily graphs can be found in the newly uploaded PDF file. Kindly refer to it for further information.

---

> > ### Comment · Reviewer_jP2E · 2023-08-13
> >
> > I've gone through the authors' responses and I have increased my initial rating and assessment since most of my previous concerns have been addressed. However, authors are still suggested to show more convincing experimental comparisons (especially digging into details of Platonov, Oleg, et al. "A critical look at the evaluation of GNNs under heterophily: are we really making progress?." ICLR 2023.) to futher enhance the technical contribution.

---

> > > ### Author Response · Authors · 2023-08-13
> > >
> > > Dear Reviewer,
> > >
> > > Thank you for reconsidering your rating and assessment based on our responses. We are glad to hear that you recognize our efforts in addressing your concerns.
> > >
> > > In response to your suggestion, we will include a description of the work by Platonov, Oleg, et al. in the Introduction section. Additionally, we will dedicate an additional section in the Appendix specifically focused on providing more experimental comparisons and visualization results using the datasets released by Platonov, Oleg, et al.
> > > We believe that incorporating these additional details and comparisons will further strengthen the technical contribution of our paper.
> > >
> > > Thank you once again for your valuable feedback. We are committed to continuously improving our work based on your suggestions.

---

### Official Review · Reviewer_mJPn · 2023-07-07

**Soundness:** 3 good
**Presentation:** 3 good
**Contribution:** 2 fair
**Rating:** 6
**Confidence:** 4

**Summary:**

Graph Neural Networks (GNNs) do not operate effectively when the graph is heterophilic with respect to the task. Addressing this issue is an important problem. The paper starts by analyzing SignedGNN model and demonstrates through a simple analysis that SignedGNNs might be implicitly implementing balance theory (at least for binary scenario). This fails in multi-class setting because e-e-f is not right. To address this issue, the aggregation is modified to remove the e-e-f aggregation, creating the weak-balanced model. However, this requires the knowledge of friends and enemies. Another model is used to predict this relationship labels. The actual model incorporates this signedGCN model along with a matrix factorization component that compares low rank behavior with the learned representations. So, the learnt representations capture low rank matrix factorization behavior while aggregating over reasonably right neighbors.

**Strengths:**

1. **Easy Approach**: The proposed approach is fairly simple.
2. **Clarity**: The paper was easy to read and follow.
3. **Good Results**: The presented results are good.

**Weaknesses:**

1. **Weak Novelty**: The proposed approach is similar to GloGNN [A]. Also, the low rank pattern has been observed in some of the prior works [B]. [C] also builds a compatibility matrix to use in their model. However, what this paper has achieved is to put these ideas together in an effective way.
2. **Unintuitive modeling choices**: The definition of $c_{ij}$ is quite unintuitive. While the individual terms have been explained, the way they are put together is inexplicable.
3. **Large number of hyperparameters**: The number of hyperparameters go up significantly in this approach. It would be useful to have a hyperparameter sensitivity study.

[A] Finding Global Homophily in Graph Neural Networks When Meeting Heterophily, ICML 2022.
[B] Simple Truncated SVD based Model for Node Classification on Heterophilic Graphs, KDD 2021 Workshop on Deep Learning on Graphs.
[C] Graph Neural Networks with Heterophily, AAAI 2021.

**Questions:**

1. While individual terms of $c_{ij}$ have been explained in the writing and some ablation study in the supplementary have been performed to show how well it works. It will still be good to understand why the three terms can be put in a very simple additive model combined with some values through Softmax.
2. It will be good to perform some hyper-parameter sensitivity study.
3. The "Impact of the Accuracy of the Estimated Signed Adjacency Matrix" was a very interesting section. It seems almost unintuitive that the model is robust to arbitrary corruption of the estimated signed adjacency matrix. However, would it not indicate that the proposed approach could also work with SignedGNN instead of WB-SignedGCN?

---

> ### Author Rebuttal · Authors · 2023-08-09
>
> Thank you for your constructive and valuable comments! Here are our responses to your concerns.
>
> **W1 The differences between LRGNN and existing GNNs.**
>
> **R1**  We put the discussion comparing LRGNN and GloGNN [1] in Appendix Sec. E, please refer to it for a detailed description. Please refer to it for a comprehensive description. Although we acknowledge that the compatibility matrix is a commonly used technique for modeling node affinity and that the low-rank pattern is not a new discovery, we firmly believe that our paper introduces novel contributions and claims. Specifically, our use of low-rank matrix completion to predict label relationship matrices and the modeling of heterophilous graphs present unique contributions. Additionally, our paper identifies limitations of existing signed GNNs and proposes a promising solution by incorporating weak balance theory. These distinct aspects of our work effectively demonstrate its novelty.
>
> **Q2 Why the three terms can be put in a very simple additive model...**
>
> **R2** The first two terms take into account the reliability of the signed adjacency matrix from varying perspectives. Intuitively, when the estimator (the pseudo label $\bar Y$) is more accurate, greater importance should be attributed to the Euclidean term. In heterophilous settings, greater importance should be assigned to the absolute term due to the effectiveness of the signed adjacency matrix generation algorithm in identifying heterophilous node pairs, as described in Section 4.2. Consequently, the significance of each term varies depending on the dataset. We leave the neural network to determine their importance using learned parameters. We visualize the learned weights using three datasets characterized by varying levels of edge homophily and estimator accuracy.
>
> |  weights  |  Wisconsin (E. H. = 0.11, Acc.=0.85) |  Cora (E. H. = 0.81, Acc.=0.87) | Squirrel (E. H. = 0.22, Acc.=0.53)|
> |  ----  | ----  | ---- |----|
> | Euclidean  |  0.38|  0.36 | 0.22 |
> | Absolute  | 0.40 |  0.34 |  0.36|
> | Attention  |    0.22|0.30 | 0.41|
>
> The table indicates that the weight assigned to the Euclidean term is slightly higher when the accuracy of the estimator is high, and the Absolute term holds great importance under heterophily settings.
>
> **Q3 It seems almost unintuitive that the model is robust to arbitrary corruption...**
>
> **R3** Our experiments show that LRGNN exhibits robustness against **Gaussian noise with a standard deviation of less than 1**. This can be attributed to the use of a square loss that provides robustness against small and dense noise, namely the Gaussian noise, as it significantly diminishes the impact of a small noise (less than 1) through squaring. In fact, the L2 norm is considered optimal when attempting to recover a matrix corrupted by Gaussian noise [4]. Further, the capped norm also effectively improves its robustness against outliers.
>
> However, this is not the case for SignedGCN. In the case of e-e-f, nodes with different labels are occasionally misclassified as belonging to the same class. This misclassification introduces sparse noise with a large value of 2, which does not align with the characteristics of "small and dense".
>
> [1]Finding Global Homophily in Graph Neural Networks When Meeting Heterophily, ICML 2022.
>
> [2]Simple Truncated SVD based Model for Node Classification on Heterophilic Graphs, KDD 2021 Workshop on Deep Learning on Graphs.
>
> [3]Graph Neural Networks with Heterophily, AAAI 2021.
>
> [4]Robust matrix factorization with unknown noise, CVPR 2013.
>
>
>
> **Q4 Concern about the number of hyperparameters**
>
> **R4** We focus on the extra hyper-parameters introduced by LRGNN, namely
> |Notation| Effect |  Search space |
> | ----| ----| ----|
> |β | weight of initial residual|  [0.5, 0.9] |
> |μ | weight of adjacency matrix for constructing the initial node representation | [0.1, 0.9]|
> |δ| tendency to generate negative edges in signed adjacency matrix| [0, 0.9] |
> |γ|  weight of the attention term in the objective function| {0.0001, 0.001, 0.002, 0} |
> |q| Operating rank| dependent on the number of classes|
> |λ | weight of L2 regularization on $U$ and $V$ |  [0.01, 0.02, 0.05, 0.001]|
> |K | number of iterations to minimize the objective function| {1, 2} |
> |Estimator| the neural network used to generate signed adjacency matrix | {GCN,MLP} |
>
> Here K and Estimator have only two choices. The Estimator can be selected based on the accuracy of these two Estimators on the designated datasets. Additionally, q can be straightforwardly set to the number of classes. As a result, there is no need to tune the hyper-parameters Estimator and q.
>
> During fine-tuning the hyper-parameters we found that our model is insensitive to the choice of K, the values of λ, and the value of γ. In fact, the selection of K and the values of λ and γ (if within the search space for λ and γ, respectively) have a very small impact on accuracy. To demonstrate, we fix these three hyper-parameters as K=1, λ=0.01, and γ=0.001, and fine-tune other hyper-parameters. The results are presented in the table below.
>
> | | Texas | Wiscon. | Cornell| Actor| Squirrel | Chamel.| Cora| Citeseer | Pubmed|
> | ----| ----| ----| ----| ----| ----| ----| ----| ----|  ----|
> |LRGNN|90.27| 88.23| 86.22| 37.34| 74.38 |79.16| 88.33| 77.53| 90.16|
> |LRGNN-restricted|88.34 $\scriptsize{\downarrow1.94}$  |82.94 $\scriptsize{\downarrow 5.29}$|85.95 $\scriptsize{\downarrow0.27}$|35.21 $\\scriptsize{\downarrow2.13}$  |72.35 $\scriptsize{\downarrow 2.03}$ |77.93 $\scriptsize{\downarrow1.23}$ |88.37$\scriptsize{\uparrow 0.04}$|76.82$\scriptsize{\downarrow{0.71}}$|  90.12$\scriptsize{\downarrow0.04}$|
>
> Here LRGNN-restricted refers to LRGNN with these three hyper-parameters being fixed and q is the number of classes. It can be observed that the decrease in accuracy is minimal, consistent with our previous statement. Thus, careful tuning is necessary only for β, μ, and δ. Tuning the hyper-parameters for LRGNN is not difficult.

---

> > ### Comment · Reviewer_mJPn · 2023-08-18
> > **Response of Reviewer mJPn**
> >
> > 1. **Limited Novelty:** I did read through the Appendix Sec. E. Regularization broadly means that the learnt variables are constrained with some known knowledge, so if the regularization happens explicitly as GloGNN or implicitly via incorporating Label Relationship predicted via the low rank structures is conceptually the same but with different implementations. The same could be said for the various other differences identified in that Section. We can think of these things as knobs that can take different values, and the current proposed approach found effective values to set, which by no means is a small feat, but novelty in terms of new ideas is still limited to incorporating the weak balance theory.
> > 2. **Combination of $c_{ij}$'s:** Again, procedurally it is quite clear what is happening, but to me Equation 13 is not a very obvious way to design it and I wanted more information on why it was designed the way it has been designed. Just to give an example, the first term's potential range is $(0, (1-1/c)^2)$, the second term's potential range is ~$(0.5, 1)$ and the third term's potential range is $(0, 1)$, so these three ranges are not the same, so it is not obvious why a simple softmax combination of these terms would even work.
> > 3. **Corruption of Adjacency:** Thanks for the clarification on this.
> > 4. **Number of hyperparameters:** It is great that some of the hyperparameters can be fixed while giving reasonable results. However, the performance on the heterophily datasets seem to vary significantly (between 1-6 percentage points). Is this because of K? or some other fixed parameter?

---

> > > ### Author Response · Authors · 2023-08-19
> > >
> > > Dear Reviewer,
> > >
> > > We sincerely appreciate your constructive and enlightening feedback. Here are our responses to your questions.
> > >
> > > **About the difference with GloGNN:** We appreciate and acknowledge your suggestion that regularization implies the constraint of learned variables based on prior knowledge. This viewpoint provides a new perspective for understanding our work. However, we would like to provide further clarification. One of the new ideas of our paper is that it is possible to find an effective way to accurately predict the majority of the unknown label relationships using the very limited known ones. The key difference lies in the amount of new information or knowledge gained.  When considering the term $||Z-A_{GCN}||$ in GloGNN and $||P_{\Omega}(UV'-A_{signed})||$ in LRGNN, the similar part of these two approaches. We can immediately find a solution for minimizing $||Z-A_{GCN}||$, namely $Z=A_{GCN}$. However, this outcome is trivial since it does not result in any new knowledge. $A_{GCN}$ is already known. Conversely, LRGNN receives a scant amount of known label relationships, say 10%, and predicts the substantial majority of the unknown ones, say 90%, which is accomplished through the decomposition of $UV'$ and the utilization of the projection function $P_\Omega$. They consider different tasks and have varying levels of complexity. $||Z-A_{GCN}||$ appears to function as an auxiliary term, enhancing the feasibility of the solution for the subspace clustering problem by approximating the adjacency matrix. From our perspective, they should not be considered as mere different implementations, as it provides a new direction to predict the unknown label relationships solely based on a small set of observations, with theoretical guarantees. The term "different implementations" might be more appropriate for describing techniques such as using L1 norm-based low-rank approximation to predict label relationships.
> > >
> > > However, we wish to emphasize that the concept of obtaining a coefficient matrix via optimization problem in GloGNN is undeniably inspiring and motivating, and we completely understand and respect that reviewers have their own criteria for a "new idea".
> > >
> > > **Combination of $c_{i,j}$'s:** Previously, we defined the first term as $\Vert \hat Y_{i,:}\Vert_2^2  \Vert \hat Y_{j,:}\Vert_2^2$, which has a range $(0,1)$. But we empirically found that $(\Vert \hat Y_{i,:}\Vert_2^2 -1/c) (\Vert \hat Y_{j,:}\Vert_2^2-1/c)$ works better, as it imposes a stronger penalty on uniformly distributed $\Vert \hat Y_{i,:}\Vert_2^2$. Regarding the combination of these three terms, our intuition is straightforward. We intend for the softmax combination to serve as an attention-like mechanism that appropriately weighs the three terms according to different datasets. But here the attention coefficients are directly parameterized ($softmax(W_a)$ where $W_a$ $\in \mathbb R^{1\times 3}$ is a randomly initialized learned matrix) rather than being computed using two involved representations as done in a standard attention function. During the early stage of the experiment, we also tried to combine these terms in alternative ways, like using $Relu(W_a)$ or  $Relu(W_a)/sum(Relu(W_a))$, as well as a more complicated edge-level scheme $[a^1_{i,j},a^2_{i,j},a^3_{i,j}]=softmax(W_a[h_i||h_j])$, where $W_a \in \mathbb R^{3 \times 2d}$ is a learned matrix. But we found the simple $softmax(W_a)$ works best as the softmax function is more attentive, and we have already included edge-level attention (the third term), so $softmax(W_a[h_i||h_j])$ may be redundant.
> > >
> > > In summary, the way of combination is motivated by the need to dynamically weigh the three terms, and after considering various alternatives, it was ultimately selected.
> > >
> > > **Large performance drop on Wisconsin dataset:** This is because of the characteristic of the dataset. As pointed out in a recent work [1], Cornell, Texas, and Wisconsin are very small (183-251 nodes), which can lead to unstable results, and the standard deviation on these datasets is very high. To mitigate this impact, we perform 100 independent runs (previously 10 runs) on these datasets to obtain more stable results.
> > >
> > > ||Texas|Wisconsin|Cornell|
> > > |----|----|----|----|
> > > |LRGNN|90.27±4.5| 88.23±3.5| 86.22±6.5|
> > > |LRGNN-restricted|88.34±3.5$\downarrow(1.93)$|86.86±4.1$\downarrow(1.37)$|85.14±5.4$\downarrow(1.08)$|
> > >
> > > Currently, the performance drop is no more than 2 percentage points. Therefore, fixing these hyperparameters would not cause a significant change in performance.
> > >
> > > Thank you once again for your valuable comments. We look forward to your reply！
> > >
> > > [1] Platonov, Oleg, et al. "A critical look at the evaluation of GNNs under heterophily: are we really making progress?." ICLR 2023.

---

> > > > ### Comment · Reviewer_mJPn · 2023-08-19
> > > > **Response of Reviewer mJPn**
> > > >
> > > > Thanks for the clarifications and repeat of experiments on the smaller datasets. Based on the above discussion, the score has been updated.

---

> > > > > ### Author Response · Authors · 2023-08-20
> > > > >
> > > > > Dear Reviewer,
> > > > >
> > > > > Thank you for your prompt and insightful feedback throughout the review process. We greatly appreciate your consideration of the above discussion and updating the score. We will keep improving our work based on the valuable insights you have provided.

---

> ### Comment · Area_Chair_xttR · 2023-08-17
> **About Authors' Reply**
>
> Dear reviewer mJPn,
>
> Would you mind to check authors' reply and indicate whether you'd like to change the score or need more discussion?
>
> AC

---

### Official Review · Reviewer_JZ4m · 2023-07-09

**Soundness:** 3 good
**Presentation:** 4 excellent
**Contribution:** 3 good
**Rating:** 7
**Confidence:** 2

**Summary:**

This paper consider predicting the global label relationship matrix as a low rank matrix completion problem. This formulation is based on the observation that the rank of the global label relationship matrix shall equal the number of classes. The authors proposed an efficient solution to the low rank matrix completion problem and the predicted global relationship matrix is used to mix features into the node embedding for predicting node labels. The authors compared the proposed method with a series of baselines and find the proposed method consistently outperform or on-par with the state-of-the-art methods.

Besides the LRGNN, the authors also prove that a variant of signed GNNs shows a tendency to follow the structural balance
72 theory, which can be enhanced by eliminating the faulty assumption from the model design.

**Strengths:**

1. The idea of computing the global label relationship matrix and then use it as a one-hop GNN is quite novel.
2. The formulation of predicting the global label relationship matrix as a signed low rank matrix completion problem is elegant.
3. The paper is extremely well-written and the logic well organized.

*I am not able to verify the math behind the matrix completion solution, but the rest of the method looks sound to me*

**Weaknesses:**

The weakness of the paper is well-discussed in the last section.

**Questions:**

It is not clear to me how the \tilde{A} entries are defined. In particular, assuming a graph has 30% of labels being known, then only roughly 9% of the edge signs are known. I am not sure how the rest of the values are determined, are they simply set to 0?

Furthermore, I am not sure how intuitively matrix completion would help determine the label relation of a node with other nodes in the graph, if the graph has no edges that has a know sign. In particular, the authors states:

*Finally, at least one observation is present per row and per column (\tilde{A}_i,i = 1)*

Yet I am not sure why that information alone would be helpful.

would appreciate if you can provide more intuition

**Limitations:**

...

---

> ### Author Rebuttal · Authors · 2023-08-09
>
> Thank you for your positive comments and for taking the time to review our paper. We appreciate your insights and would like to address your concerns.
>
> First, regarding our optimization algorithm for the matrix completion problem, we would like to provide some additional information.
>
> The optimization algorithm we used follows the approach outlined in a highly regarded paper [1] and we have properly cited and acknowledged this source in our paper. The Majorization-Minimization technique, upon which our algorithm is based, is elegant and conceptually straightforward. Its correctness is easily verifiable. We have put significant effort into verifying the mathematical aspects of this algorithm to ensure its correctness. We would love to describe the basic operations of the Majorization-Minimization to make sure that you would not have concerns about the correctness of the algorithm.
>
> Consider we wish to solve $\underset { x}{min} f(x)$ that is so **complicated** that we cannot handle the problem directly. We aim to find a sequence $\left\\{x_k\right\\}$ such that $f(x_{k+1})\le f(x_{k})$. The idea is about using the variable x at the current iteration ($x_k$) to construct a **simpler** surrogate function $g(x|x_k)$, so we can obtain $x_{k+1}$ by solving $x_{k+1} = \underset { x}{arg min} \\; g(x|x_k)$, where $g(x|x_k)$ can be any function once it satisfies the following two conditions:
>
>  **(1)** $f(x) \le g(x|x_k)$, $\forall x$
>
> **(2)** $f(x_k) = g(x_k|x_k)$, $\forall x_k$
>
> We can easily prove $f(x_{k+1}) \underset {(1)} {\le} g(x_{k+1}|x_k) \underset {\text{by definition}}{\le} g(x_k|x_k) \underset {(2)}{=} f(x_k)$. The intuition is to minimize $f$ by another simpler function $g$ such that minimizing $g$ ’helps’ minimize $f$.
>
> Now back to our paper, the construction of the surrogate function is described by Eq.(24) (page 22), and the proof of the two conditions above refers to Eq.(37),(38), and (39) (page 24).
>
> **Q1** I am not sure how the rest of the values are determined, are they simply set to 0?
>
> **R1** Yes! We just simply set the unknown entries to 0, but you can set them to any other specific values if you like. The rest values of $\tilde{A}$ do not affect the objective function or the values of $U$ and $V$ because only the known entries contribute to the objective function Eq.(8). Specifically, the first term of the objective function is given by $\sum_{(i,j)\in \mathcal{E}} (UV^T_{i,j}-\tilde{A}_{i,j})$. Here, $\mathcal{E}$ represents the observation set, which includes the known entries.
>
> **Q2** Finally, at least one observation is present per row and per column (\tilde{A}_i,i = 1). Why that information alone would be helpful.
>
> **R2**  This is only a consideration at the theoretical level. Look at the objective function again.
>
> $O= \sum_{(i,j)\in \mathcal{E}} (UV^T_{i,j}-\tilde{A}_{i,j})$
>
> If there are no observed entries in a specific row, such as row 1 with $(1,j)\notin \mathcal{E}, \forall j$, the value of $U_{1,:}$ does not affect $O$. Consequently, the value of $U_{1,:}$ remains unchanged throughout the iteration process, as it does not contribute to the gradient. Thus, its value is solely determined by the initialization method and random seed employed. Knowing $\tilde A_{i,i} = 1$ alone is **insufficient** to infer its label relationships by matrix completion, but at least we can generate a unique $U_{i,:}$ and $V_{i,:}$ that are independent of the initialization method and the random seed. For other cases demonstrating how the label relation of a node with other nodes can aid in matrix completion, please refer to Appendix Section A.
>
> Thanks for your time and effort again! Please let us know if you have any further question.
>
> [1] Hastie, Trevor, et al. "Matrix completion and low-rank SVD via fast alternating least squares." The Journal of Machine Learning Research 16.1 (2015): 3367-3402.

---

> ### Comment · Area_Chair_xttR · 2023-08-17
> **About Authors' Reply**
>
> Dear reviewer JZ4m,
>
> Would you mind to check authors' reply and indicate whether you'd like to change the score or need more discussion?
>
> AC

---

### Official Review · Reviewer_rmTT · 2023-07-20

**Soundness:** 2 fair
**Presentation:** 3 good
**Contribution:** 2 fair
**Rating:** 5
**Confidence:** 5

**Summary:**

This paper introduces the LRGNN method, which aims to enhance the performance on heterophilous graphs by re-constructing the signed relationships. The approach involves optimizing the low-rank signed matrix using the SVD-free LRMF strategy.


**Strengths:**

The utilization of the low-rank signed relationship matrix in heterophilous graph representation is intriguing and promising.


**Weaknesses:**

Weaknesses:

1. The limitations on the first contribution.
One of the limitations of this method is its reliance on the data split. The success of low-rank matrix completion is highly dependent on the availability and quality of the signed adjacency matrix and the observation set. In cases where the observation set is small, there may be nodes that have no neighbors selected, as mentioned in Line 140. This poses challenges in effectively guessing the values of U and V, potentially impacting the accuracy of the low-rank matrix completion process.

2. Limited novelty:
Although the method incorporates low-rank optimization, the overall novelty of the approach is somewhat limited. The designed method in Section 4 shares similarities with other existing methods. For instance, when compared to the general compatibility matrix used in [1,2], the primary difference lies in both matrices' objective to measure whether two nodes belong to the same class.

3. Concerns with the experiments:
a. Effectiveness of low-rank recovery of Z (Eq. 7 and 10):
To further justify the effectiveness of the low-rank recovery of Z, it is essential to provide evaluations for Eq. (7) and (10). Demonstrating the accuracy and reliability of the low-rank recovery process would strengthen the paper's claims.
b. Parameter numbers:  Including the parameter numbers in the paper would provide valuable insights, especially regarding the growth of the matrices U and V in correlation with the number of nodes. This information would contribute to a more comprehensive understanding of the method's scalability and resource requirements.


Detailed comments:
1. Fig.1: In panel t3, the color of the circle appears to be incorrect. Please ensure that the color
corresponds accurately to the represented class.

2. Fig.4 Wisconsin: There seems to be a discrepancy in the edge numbers of in-class and between-class connections. Given that this dataset is heterophilous, the number of edges within the same class and between different classes should differ significantly. Please review and verify the edge counts for accuracy.

[1] Graph Neural Networks with Heterophily.
[2] Powerful Graph Convolutioal Networks with Adaptive Propagation Mechanism for Homophily and Heterophily.

**Questions:**

Please check the weakness

**Limitations:**

Please check the weakness

---

> ### Author Rebuttal · Authors · 2023-08-09
>
> Thanks for the detailed review. We address the concerns in below.
>
> **Q1: The reliance on the size of the observation set**
>
> **R** In order to align with the time complexity of other GNNs, we define the observation set as the edge set. However, it is important to note that the performance of the GNN algorithm is known to be influenced by the number of edges present in the graph. So if a small observation set poses a challenge to the low-rank approximation method, it would be also a challenge to GNN models, since they use the same edge set. There is no evidence suggesting that the low-rank approximation method is more sensitive to this size than a GNN model. In fact, our experimental results demonstrate that our LRGNN achieves the best or runner-up results on sparse graphs such as the Texas, Wisconsin, and Cornell datasets. Furthermore, the visualization results presented in Figures 4 and 5 showcase that low-rank approximation can recover good label relationship matrices even if the edges are sparse.
> Additionally, there is no difference between low-ran approximation and GNNs when encountering isolated nodes. For instance, when a node contains no links, the output representation of this node by a GNN model is equivalent to that of its MLP counterpart. Similarly, the corresponding row of the node in the matrix recovered by a low-rank approximation model has only one non-zero element, then using it for propagation we also obtain the same outcome as an MLP.
>
> Since all types of GNNs are influenced by the number of edges and there is empirical evidence that low-rank matrix completion performs effectively with limited observation sets, this should not be considered a particular drawback of our proposed approach or paper.
>
> **Q2:  the reliance on the quality of the signed adjacency matrix**
>
> **R** Suppose the accurate signed adjacency matrix is given by $A_{ground}$, then the estimated signed adjacency matrix can be expressed as $\tilde A = A_{ground} + N$, where $N$ is a noise matrix.  We have dedicated much effort to exploring the influence of the noise in Appendix B, including the effectiveness of the capped norm and the change in performance of LRGNN when the entries of $N$ are i.i.d. drawn from Gaussian distribution.
>
> For example, LRGNN outperforms the runner-up OGNN by a substantial margin on the Squirrel dataset. It is worth noting that the signed adjacency matrix of Squirrel is particularly inaccurate, as reflected by the estimator's low accuracy of approximately 50%. There is also a visualization result of the recovered $Z$ on Squirrel presented on page 21, figure 14(b). These results suggest that the quality of the generated signed adjacency matrix has a very slight impact on LRGNN’s performance, and the outstanding performance of LRGNN is not conditioned on a very accurate signed adjacency matrix.
>
> **Q3: limited novelty**
>
> **R** While compatibility matrices are commonly employed to model node affinity, our paper delves into several innovative aspects that set it apart. Firstly, we introduce low-rank matrix completion to the GNNs, showcasing its power in predicting the label relationship matrix.  Secondly, we establish that predominant signed GNNs find their foundation in balance theory. Our empirical analysis further highlights that these can be significantly enhanced by reconsidering the 'e-e-f' assumption, which might not always hold true.  Notably, the reviewer has also positively remarked on our approach, mentioning, "The exploitation of the low-rank signed relationship matrix in heterophilous graph representation is both intriguing and holds promise." This attests to the novelty we're bringing into the domain. In summation, our primary contributions and the claims made in the paper distinctively address areas that have been relatively untouched in preceding research, thereby underscoring the paper's innovative value.
>
> **Q4 Effectiveness of low-rank recovery of Z (Eq. 7 and 10)**
>
> **R** We would like to clarify a few points. Firstly, Eq. 7 pertains to a different aspect of our methodology and does not directly address the recovery of Z. Conversely, Eq. 10 is instrumental in generating the signed adjacency matrix. As for the low-rank recovery's precision, our study showcases a plethora of visualization results underscoring its effectiveness. These include the recovery error rate on real-world graphs illustrated in Figure 3, visualization of the recovered Z in Figures 4 and 5, and recovery accuracy on generated data with a minimal observation rate presented in Figures 7 and 8 (Appendix A). We invite you to review these figures for a more comprehensive understanding.
>
> **Q5 Parameter numbers**
>
> **R** U and V are not parameterized, and they are expressed as a series of matrix multiplication, see Eq. 25 and Eq. 27 for their formulation. Regarding the size of U and V, they are of size $n \times q$, where q is a small number typically close to the # of classes. Therefore, U and V would not require more computational or storage resources than a typical node representation matrix in a GNN model. The parameter number of LRGNN is $ O(fd+dc+nd) $, where $f$ is the dimension of the raw feature, $d$ is the hidden size, $n$ the number of nodes. As a comparison, the parameter number of GCN is $O(fd+dc+d^2)$, and the additional parameters of LRGNN is only linear to the number of nodes.
>
> **Q6 Fig.1 & Fig. 4**
>
> **R** We will correct that mistake in Fig 1. Note that Figure 4 is the visualization of $Z$ with $n \times n$ non-zero entries, which can be viewed as the predicted adjacency matrix of a signed **complete graph**, not the sparse adjacency matrix. Therefore, the number of edges within the same class and between different classes is only related to the class balance, irrespective of heterophily or homophily. Fig.4 Wisconsin is accurate.

---

> > ### Comment · Reviewer_rmTT · 2023-08-17
> >
> > I have read the reply and the questions have been addressed. I will change my score from 4 to 5.

---

> > > ### Author Response · Authors · 2023-08-17
> > >
> > > Dear Reviewer,
> > >
> > > Thank you for taking the time to read our response. We sincerely appreciate your insightful feedback and adjustment in your score. We will keep improving our paper based on your suggestions.

---

> ### Comment · Area_Chair_xttR · 2023-08-17
> **About Authors' Reply**
>
> Dear reviewer rmTT,
>
> Would you mind to check authors' reply and indicate whether you'd like to change the score or need more discussion?
>
> AC

---

### Author Rebuttal · Authors · 2023-08-07

Dear Reviewers and AC,

In response to the concerns raised by Reviewer jP2E, we have presented tables that juxtapose LRGNN with existing baselines across a spectrum of recent and large-scale datasets. This should help alleviate concerns regarding the scope of datasets used in our initial study. To elucidate on the distinctions, we delve deeper into the expressive constraints of FAGCN. One of the core tenets of our work is the novel integration of low-rank matrix completion within GNNs, particularly tailored for Heterophily graphs. Alongside this, our paper also sheds both theoretical and empirical light on the inherent limitations of the current crop of signed GNNs – an exploration that remains largely untouched in prior research. Addressing the queries posed by Reviewer mJPn, we have included a hyper-parameter sensitivity analysis. Furthermore, to offer a transparent view on our methodology, a table elucidating the learned weights has been incorporated. This should provide clarity on the amalgamation of terms in $c_{i,j}$ and any apprehensions regarding the volume of hyperparameters. For reviewer rmTT, who expressed concerns about the potential dependency on the size and quality of the observed entries, we direct attention to the plethora of figures and examples (both from real-world scenarios and generated datasets) in Appendix A and B of our initial submission, underscoring the resilience of our approach. Lastly, we have taken steps to clearly address the questions from Reviewer JZ4m, elucidating on the nature of $\tilde{A}$, diving deeper into matrix completion, and providing a clearer perspective on our optimization algorithm.

We want to express our gratitude to the reviewers for their valuable suggestions and would greatly appreciate any further questions they may have regarding our paper. If our responses have addressed your questions, we would kindly ask for your reconsideration of the scores.

---

### Decision · Program_Chairs · 2023-09-21

**Decision:**

Accept (poster)

**Comment:**

**Summary:**

The paper proposes a low-rank graph neural network (LRGNN) that predicts the global label relationship matrix using a low-rank matrix completion technique. The predicted matrix is then used to aggregate node features in a GNN framework. The paper provides theoretical and empirical evidence for the effectiveness and robustness of LRGNN on various real-world and synthetic datasets.

**Strengths:**
- The paper presents a novel and elegant approach that combines low-rank matrix completion and GNNs to model the label relationships in heterophilous graphs.
- The paper is well-written, clear, and easy to follow.

**Weaknesses:**
- The novelty of LRGNN is somewhat limited, as it builds upon existing techniques such as compatibility matrices, low-rank matrix factorization, and signed GNNs.
- The number of hyperparameters in LRGNN is large, which may affect its scalability and generalization.

**Decision:**

After the rebuttal, all reviewers are generally positive towards this submission. An acceptance (poster) is suggested.